# WHAT CAN BE LEARNT WITH WIDE CONVOLUTIONAL NEURAL NETWORKS?

## ABSTRACT

Understanding how convolutional neural networks (CNNs) can efficiently learn high-dimensional functions remains a fundamental challenge. A popular belief is that these models harness the local and hierarchical structure of natural data such as images. Yet, we lack a quantitative understanding of how such structure affects performance, e.g. the rate of decay of the generalisation error with the number of training samples. In this paper, we study deep CNNs in the kernel regime. First, we show that the spectrum of the corresponding kernel inherits the hierarchical structure of the network, and we characterise its asymptotics. Then, we use this result together with generalisation bounds to prove that deep CNNs adapt to the spatial scale of the target function. In particular, we find that if the target function depends on low-dimensional subsets of adjacent input variables, then the rate of decay of the error is controlled by the effective dimensionality of these subsets. Conversely, if the target function depends on the full set of input variables, then the error rate is inversely proportional to the input dimension. We conclude by computing the rate when a deep CNN is trained on the output of another deep CNN with randomly-initialised parameters. Interestingly, we find that, despite their hierarchical structure, the functions generated by deep CNNs are too rich to be efficiently learnable in high dimension.

## 1 INTRODUCTION

Deep convolutional neural networks (CNNs) are particularly successful in certain tasks such as image classification. Such tasks generally entail the approximation of functions of a large number of variables, for instance the number of pixels which determine the content of an image. Learning a generic high-dimensional function is plagued by the *curse of dimensionality*: the rate at which the generalisation error $\epsilon$ decays with the number of training samples $n$ vanishes as the dimensionality $d$ of the input space grows, i.e. $\epsilon(n) \sim n^{-\beta}$ with $\beta = O(1/d)$ (Wainwright, 2019). Therefore, the success of CNNs in classifying data whose dimension can be in the hundreds or more (Hestness et al., 2017; Spigler et al., 2020) points to the existence of some underlying structure in the task that CNNs can leverage. Understanding the structure of learnable tasks is arguably one of the most fundamental problems in deep learning, and also one of central practical importance—as it determines how many examples are required to learn up to a certain error. A popular hypothesis is that learnable tasks are local and hierarchical: features at any scale are made of sub-features of smaller scales. Although many works have investigated this hypothesis (Biederman, 1987; Poggio et al., 2017; Kondor & Trivedi, 2018; Zhou et al., 2018; Deza et al., 2020; Kohler et al., 2020; Poggio et al., 2020; Schmidt-Hieber, 2020; Finocchio & Schmidt-Hieber, 2021; Giordano et al., 2022), there are no available predictions for the exponent $\beta$ for deep CNNs trained on tasks with a varying degree of locality or a truly hierarchical structure.

In this paper we perform such a computation in the overparameterised regime, where the width of the hidden layer of the neural networks diverges and the network output is rescaled so as to converge to that of a kernel method (Jacot et al., 2018; Lee et al., 2019). Although the deep networks deployed in real scenarios do not generally operate in such regime, the connection with the theory of kernel regression provides a recipe for computing the decay of the generalisation error with the number of training examples. Namely, given an infinitely wide neural network, its generalisation abilities depend on the spectrum of the corresponding kernel (Caponnetto & De Vito, 2007; Bordelon et al.,

2020): the main challenge is then to characterise this spectrum, especially for deep CNNs whose kernels are rather cumbersome and defined recursively (Arora et al., 2019). This characterisation is the main result of our paper, together with the ensuing study of generalisation in deep CNNs.

## 1.1 OUR CONTRIBUTIONS

More specifically, this paper studies the generalisation properties of deep CNNs with non-overlapping patches and no pooling (defined in Sec. 2, see Fig. 1 for an illustration), trained on a target function $f^*$ by empirical minimisation of the mean squared loss. We consider the infinite-width limit (Sec. 3) where the model parameters change infinitesimally over training, thus the trained network coincides with the predictor of kernel regression with the Neural Tangent Kernel (NTK) of the network. Due to the equivalence with kernel methods, generalisation is fully characterised by the spectrum of the integral operator of the kernel: in simple terms, the projections on the eigenfunctions with larger eigenvalues can be learnt (up to a fixed generalisation error) with fewer training points (see, e.g., Bach (2021)).

**Spectrum of deep hierarchical kernels (Thm. 3.1).** Due to the network architecture, the hidden neurons of each layer depend only on a subset of the input variables, known as the receptive field of that neuron (highlighted by coloured boxes in Fig. 1, left panel). We find that the eigenfunctions of the NTK of a hierarchical CNN of depth $L + 1$ can be organised into sectors $l = 1, \ldots, L$ associated with the hidden layers of the network (Thm. 3.1). The eigenfunctions of each sector depend only on the receptive fields of the neurons of the corresponding hidden layer: if we denote with $d_{\text{eff}}(l)$ the size of the receptive fields of neurons in the $l$-th hidden layer, then the eigenfunctions of the $l$-th sector are effectively functions of $d_{\text{eff}}(l)$ variables. We characterise the asymptotic behaviour of the NTK eigenvalues with the degree of the corresponding eigenfunctions (Thm. 3.1) and find that it is controlled by $d_{\text{eff}}(l)$. As a consequence, the eigenfunctions with the largest eigenvalues—the easiest to learn—are those which depend on small subsets of the input variables and have low polynomial degree. This is our main technical contribution and all of our conclusions follow from it.

**Adaptivity to the spatial structure of the target (Cor. 4.1).** We use the above result to prove that deep CNNs can adapt to the spatial scale of the target function (Sec. 4). More specifically, by using rigorous bounds from the theory of kernel ridge regression (Caponnetto & De Vito, 2007) (reviewed in the first paragraph of Sec. 4), we show that when learning with the kernel of a CNN and optimal regularisation, the decay of the error depends on the effective dimensionality of the target $f^*$—i.e., if $f^*$ only depends on $d_{\text{eff}}$ adjacent coordinates of the $d$-dimensional input, then $\epsilon \sim n^{-\beta}$ with $\beta \geq O(1/d_{\text{eff}})$ (Cor. 4.1, see Fig. 1 for a pictorial representation). We find a similar picture in ridgeless regression by using non-rigorous results derived with the replica method (Bordelon et al., 2020; Loureiro et al., 2021) (Sec. 5). Notice that for targets which are spatially localised (or sums of spatially localised functions), the rates achieved with deep CNNs are much closer to the Bayes-optimal rates—realised when the architecture is fine-tuned to the structure of the target—than $\beta = O(1/d)$ obtained with the kernel of a fully-connected network. Moreover, we find that hierarchical functions generated by the output of deep CNNs are too rich to be efficiently learnable in high dimensions (Lemma 5.1). We confirm these results through extensive numerical studies and find them to hold even if the nonoverlapping patches assumption is relaxed (Subsec. G.4).

## 1.2 RELATED WORK

The benefits of shallow CNNs in the kernel regime have been investigated by Bietti (2022); Favero et al. (2021); Misiakiewicz & Mei (2021); Xiao & Pennington (2022); Xiao (2022); Geifman et al. (2022). Favero et al. (2021), and later (Misiakiewicz & Mei, 2021; Xiao & Pennington, 2022), studied generalisation properties of shallow CNNs, finding that they are able to beat the curse of dimensionality on local target functions. However, these architectures can only approximate functions of single input patches or linear combinations thereof. Bietti (2022), in addition, includes generic pooling layers and begins considering the role of depth by studying the approximation properties of kernels which are integer powers of other kernels. We generalise this line of work by studying CNNs of any depth with nonanalytic (ReLU) activations: we find that the depth and nonanalyticity of the resulting kernel are crucial for understanding the inductive bias of deep CNNs. This result should also be contrasted with the spectrum of the kernels of deep fully-connected networks, whose asymptotics

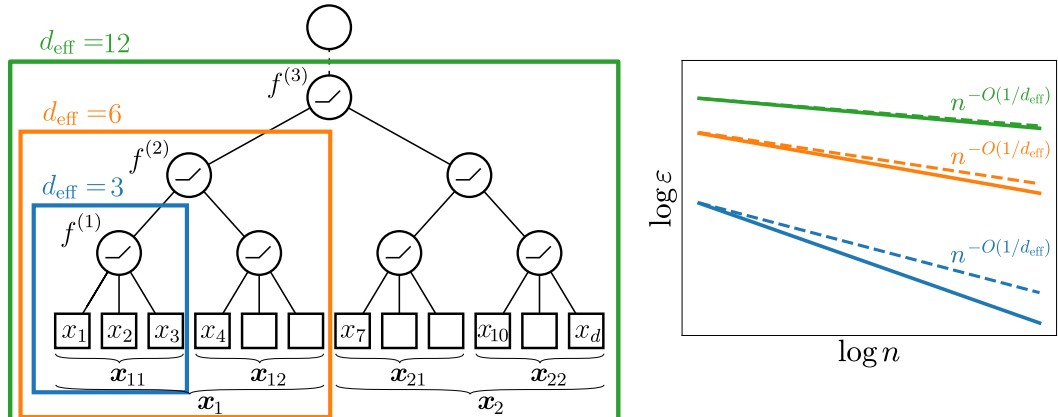

Figure 1: **Left:** Computational skeleton of a convolutional neural network of depth $L+1=4$ ($L=3$ hidden layers). The leaves of the graph (squares) correspond to input coordinates, and the root (empty circle) to the output. All other nodes represent (infinitely wide layers of) hidden neurons. We define as 'meta-patches' (i.e. patches of patches) the sets of input variables that share a common ancestor node along the tree (such as the squares within each coloured rectangle). Each meta-patch coincides with the receptive field of the neuron represented by this common ancestor node, as indicated below the input coordinates. For each hidden layer $l = 1, \ldots, L$, there is a family of meta-patches having dimensionality $d_{\text{eff}}(l)$. **Right:** Sketches of learning curves $\epsilon(n)$ obtained by learning target functions of varying spatial scale with the network on the left. More specifically, the target is a function of a 3-dimensional patch for the blue curve, a 6-dimensional patch for the orange curve, and the full input for the green curve. We predict (and confirm empirically) that both the decay of $\epsilon$ with $n$ (full lines) and the rigorous upper bound (dashed lines) are controlled by the effective dimensionality of the target.

do not depend on depth (Bietti & Bach, 2021). Furthermore, we extend the analysis of generalisation to target functions that have a hierarchical structure similar to that of the networks themselves.

Geifman et al. (2022) derive bounds on the spectrum of the kernels of deep CNNs. However, they consider only filters of size one in the first layer and do not include a theoretical analysis of generalisation. Instead, we allow filters of general dimension and give tight estimates of the asymptotic behaviour of eigenvalues, which allow us to predict generalisation properties. Xiao (2022) is the closest to our work, as it also investigates the spectral bias of deep CNNs in the kernel regime. However, it considers a different limit where both the input dimension and the number of training points diverge and does not characterise the asymptotic decay of generalisation error with the number of training samples.

Paccolat et al. (2021); Malach & Shalev-Shwartz (2021); Abbe et al. (2022) use sparse target functions which depend only on a few of the input variables to prove sample complexity separation results between networks operating in the kernel regime and in the feature regime—where the change in parameters during training can be arbitrarily large. In this respect, our work shows that when the few relevant input variables are adjacent, i.e. the target function is spatially localised, deep CNNs achieve near-optimal performances even in the kernel regime.

## 2 NOTATION AND SETUP

Our work considers CNNs with nonoverlapping patches and no pooling layers. These networks are fully characterised by the depth $L + 1$ (or number of hidden layers $L$) and a set of filters sizes $\{s_l\}_l$ (one per hidden layer). We call such networks *hierarchical* CNNs.

**Definition 2.1 ($L$-hidden-layers hierarchical CNN)** *Denote by $\sigma$ the normalised ReLU function, $\sigma(x) = \sqrt{2}\max(0, x)$. For each input $\boldsymbol{x} \in \mathbb{R}^{d\,1}$ and $s$ a divisor of $d$, denote by $\boldsymbol{x}_i$ the $i$-th*

---

[1]Notice that all our results can be readily extended to image-like input signals $\{x_{ij}\}_{i,j}$ or tensorial objects with an arbitrary number of indices.

$s$-dimensional patch of $\boldsymbol{x}$, $\boldsymbol{x}_i = (x_{(i-1) \times s+1}, \dots, x_{i \times s})$ for all $i = 1, \dots, d/s$. The output of a $L$-hidden-layers hierarchical neural network can be defined recursively as follows.

$$f_{h,i}^{(1)}(\boldsymbol{x}) = \sigma\left(\boldsymbol{w}_h^{(1)} \cdot \boldsymbol{x}_i\right), \ \forall h \in [1 \, .. \, H_1], \ \forall i \in [1 \, .. \, p_1];$$

$$f_{h,i}^{(l)}(\boldsymbol{x}) = \sigma\left(\frac{1}{\sqrt{H_{l-1}}} \sum_{h'} \frac{\boldsymbol{w}_{h,h'}^{(l)} \cdot \left(\boldsymbol{f}_{h'}^{(l-1)}\right)_i}{\sqrt{s_l}}\right), \ \forall h \in [1 \, .. \, H_l], \ i \in [1 \, .. \, p_l], \ l \in [2 \, .. \, L];$$

$$f(\boldsymbol{x}) = f^{(L+1)}(\boldsymbol{x}) = \frac{1}{\sqrt{H_L}} \sum_{h=1}^{H_L} \sum_{i=1}^{p_L} \frac{w_{h,i}^{(L+1)} f_{h,i}^{(L)}(\boldsymbol{x})}{\sqrt{p_L}}. \tag{1}$$

$H_l$ denotes the width of the $l$-th layer, $s_l$ the filter size ($s_1 = s$), $p_l$ the number of patches ($p_1 \equiv p = d/s$). $\boldsymbol{w}_h^{(1)} \in \mathbb{R}^{s_1}$, $\boldsymbol{w}_{h,h'}^{(l)} \in \mathbb{R}^{s_l}$, $w_{h,i}^{(L+1)} \in \mathbb{R}$.

Hierarchical CNNs are best visualised by considering their computational skeleton, i.e. the directed acyclic graph obtained by setting $H_l = 1 \ \forall \ l$ (example in Fig. 1, left, with $L = 3$ hidden layers and filter sizes $(s_1, s_2, s_3) = (3, 2, 2)$). Having nonoverlapping patches, the computational skeleton is an ordered tree, whose root is the output (empty circle at the top of the figure) and the leaves are the input coordinates (squares at the bottom). All the other nodes represent neurons and all the neurons belonging to the same hidden layer have the same distance from the input nodes. The tree structure highlights that the post-activations $f_i^l$ of the $l$-th layer depend only on a subset of the input variables, also known as the *receptive field*.

Since the first layer of a hierarchical CNN acts on $s_1$-dimensional patches of the input, it is convenient to consider each $d$-dimensional input signal as the concatenation of $p$ $s$-dimensional patches, with $s = s_1$ and $p \times s = d$. We assume that each patch is normalised to $1$ [2], so that the input space is a product of $p$ $s$-dimensional unit spheres (called multisphere in Geifman et al. (2022)):

$$\mathsf{M}^p \mathbb{S}^{s-1} := \prod_{i=1}^p \mathbb{S}^{s-1} \subset \mathbb{S}^{d-1}. \tag{2}$$

Notice that the $s$-dimensional patches are also the receptive fields of the first-hidden-layer neurons (as in the blue rectangle in Fig. 1 for $s = 3$). In general, the receptive field of a neuron in the $l$-th hidden layer with $l > 1$ is a group of $\prod_{l'=2}^l s_{l'}$ adjacent patches (as in the orange rectangle of Fig. 1 for $l = 2$, $s_2 = 2$ or the green rectangle for $l = 3$, $s_3 = s_2 = 2$), which we refer to as a *meta-patch*. Due to the correspondence with the receptive fields, each meta-patch is identified with one path on the computational skeleton: the path which connects the output node to the hidden neuron whose receptive field coincides with the meta-patch. If such hidden neuron belongs to the $l$-th hidden layer, the path is specified by a tuple of $L - l + 1$ indices, $i_{l+1 \to L+1} := i_{L+1} \dots i_{l+1}$, where each index indicates which branch to select when descending from the root to the neuron node. With this notation, $\boldsymbol{x}_{i_{l+1 \to i_{L+1}}}$ denotes one of the $p_l$ meta-patches of size $\prod_{l' \leq l} s_{l'}$. Because of the normalisation of the $s_1$-dimensional patches, each meta-patch has an *effective dimensionality* which is lower than its size:

$$\boldsymbol{x}_{i_{2 \to L+1}} \in \mathbb{S}^{s_1 - 1} \Rightarrow \begin{cases} d_{\text{eff}}(1) := \dim(\boldsymbol{x}_{i_{2 \to L+1}}) = (s_1 - 1), \\ d_{\text{eff}}(l) := \dim(\boldsymbol{x}_{i_{l+1 \to L+1}}) = (s_1 - 1)\prod_{l'=2}^l s_{l'}, \quad \forall l \in [2 \, .. \, L]. \end{cases} \tag{3}$$

## 3 HIERARCHICAL KERNELS AND THEIR SPECTRA

We turn now to the infinite-width limit $H_l \to \infty$: because of the aforementioned equivalence with kernel methods, this limit allows us to deduce the generalisation properties of the network from the spectrum of a kernel. In this section, we present the kernels corresponding to the hierarchical models of Def. 2.1 and characterise the spectra of the associated integral operators.

We consider specifically two kernels: the *Neural Tangent Kernel* (NTK), corresponding to training all the network parameters (Jacot et al., 2018); and the *Random Feature Kernel* (RFK), corresponding to training only the weights of the linear output layer (Rahimi & Recht, 2007; Daniely et al.,

---

[2]We show in Subsec. G.4 that our predictions remain true if the inputs are sampled uniformly in the $d$-dimensional hypercube $[0, 1]^d$ or from a Gaussian distribution on $\mathbb{R}^d$.

2016). In both cases, the kernel reads:

$$\mathcal{K}(\boldsymbol{x}, \boldsymbol{y}) = \sum_{\text{trained params } \theta} \partial_\theta f(\boldsymbol{x}) \partial_\theta f(\boldsymbol{y}). \tag{4}$$

The NTK and RFK of deep CNNs have been derived previously by Arora et al. (2019). In App. B we report the functional forms of these kernels in the case of hierarchical CNNs. These kernels inherit the hierarchical structure of the original architecture and their operations can be visualised again via the tree graph of Fig. 1. In this case, the leaves represent products between the corresponding elements of two inputs $\boldsymbol{x}$ and $\boldsymbol{y}$., i.e. $x_1 y_1$ to $x_d y_d$, and the root the kernel output $\mathcal{K}(\boldsymbol{x}, \boldsymbol{y})$. The output can be built layer by layer by following the same recipe for each node: first sum the outputs of the previous layer which are connected to the present node, then apply some nonlinear function which depends on the activation function of the network. In particular, for each couple of inputs $\boldsymbol{x}$ and $\boldsymbol{y}$ on the multisphere $\mathsf{M}^p \mathbb{S}^{s-1}$, hierarchical kernels depend on $\boldsymbol{x}$ and $\boldsymbol{y}$ via the $p$ dot products between corresponding $s$-dimensional patches of $\boldsymbol{x}$ and $\boldsymbol{y}$. As a comparison, Bietti & Bach (2021) showed that the NTK and RFK of a fully-connected network of any depth depend on the full dot product $\boldsymbol{x} \cdot \boldsymbol{y}$, whereas those of a shallow CNN can be written as the sum of $p$ kernels, each depending on only one of the patch dot products (Favero et al., 2021).

Given the kernel, the associated integral operator reads

$$(T_{\mathcal{K}} f)(\boldsymbol{x}) := \int_{\mathbb{S}^{s-1}} \mathcal{K}(\boldsymbol{x}, \boldsymbol{y}) f(\boldsymbol{y}) dp(\boldsymbol{y}), \tag{5}$$

with $dp(\boldsymbol{x})$ denoting the uniform distribution of input points on the multisphere. The spectrum of this operator provides, via Mercer's theorem (Mercer, 1909), an alternative representation of the kernel $\mathcal{K}(\boldsymbol{x}, \boldsymbol{y})$ and a basis for the space of functions that the kernel can approximate. The asymptotic decay of the eigenvalues, in particular, is crucial for the generalisation properties of the kernel, as it will be clarified at in Sec. 4. Since the input space is a product of $s$-dimensional unit spheres and the kernel depends on the $p$ scalar products between corresponding $s$-dimensional patches of $\boldsymbol{x}$ and $\boldsymbol{y}$, the eigenfunctions of $T_{\mathcal{K}}$ are products of spherical harmonics acting on the patches (see App. A for definitions and the relevant background). For the sake of clarity, we limit the discussion in the main paper to the case $s = 2$, where, since each patch $\boldsymbol{x}_i$ is entirely determined by an angle $\theta_i$, the multisphere $\mathsf{M}^p \mathbb{S}^{s-1}$ reduces to the $p$-dimensional torus and the eigenfunctions to $p$-dimensional plane waves: $e^{i\boldsymbol{k} \cdot \boldsymbol{\theta}}$ with $\boldsymbol{\theta} := (\theta_1, \dots, \theta_p)$ and label $\boldsymbol{k} := (k_1, \dots, k_p)$. In this case the eigenvalues coincide with the $p$-dimensional Fourier transform of the kernel $\mathcal{K}(\cos\theta_1, \dots, \cos\theta_p)$ and the large-$\boldsymbol{k}$ asymptotics are controlled by the nonanalyticities of the kernel (Widom, 1963). The general case with patches of arbitrary dimension is presented in the appendix.

**Theorem 3.1 (Spectrum of hierarchical kernels)** *Let $T_{\mathcal{K}}$ be the integral operator associated with a $d$-dimensional hierarchical kernel of depth $L + 1$, $L > 1$ and filter sizes $(s_1, \dots, s_L)$ with $s_1 = 2$. Eigenvalues and eigenfunctions of $T_{\mathcal{K}}$ can be organised into $L$ sectors associated with the hidden layers of the kernel/network. For each $1 \leq l \leq L$, the $l$-th sector consists of $(\prod_{l'=1}^{l} s_{l'})$-local eigenfunctions: functions of a single meta-patch $\boldsymbol{x}_{i_{l+1 \to L+1}}$ which cannot be written as linear combinations of functions of smaller meta-patches. The labels $\boldsymbol{k}$ of these eigenfunctions are such that there is a meta-patch $\boldsymbol{k}_{i_{l+1 \to L+1}}$ of $\boldsymbol{k}$ with no vanishing sub-meta-patches and all the $k_i$'s outside of $\boldsymbol{k}_{i_{l+1 \to L+1}}$ are $0$ (because the eigenfunction is constant outside of $\boldsymbol{x}_{i_{l+1 \to L+1}}$). The corresponding eigenvalue is degenerate with respect to the location of the meta-patch: we call it $\Lambda_{\boldsymbol{k}_{i_{l+1 \to i_{L+1}}}}^{(l)}$.*

*When $\|\boldsymbol{k}_{i_{l+1 \to L+1}}\| \to \infty$, with $k = \|\boldsymbol{k}_{i_{l+1 \to L+1}}\|$,*

$$\Lambda_{\boldsymbol{k}_{i_{l+1 \to L+1}}}^{(l)} = \mathcal{C}_{2,l} \, k^{-2\nu - d_{\text{eff}}(l)} + o\left(k^{-2\nu - d_{\text{eff}}(l)}\right), \tag{6}$$

*with $\nu_{\text{NTK}} = 1/2$, $\nu_{\text{RFK}} = 3/2$ and $d_{\text{eff}}$ the effective dimensionality of the meta-patches defined in Eq. 3. $\mathcal{C}_{2,l}$ is a strictly positive constant for $l \geq 2$ whereas for $l = 1$ it can take two distinct strictly positive values depending on the parity of $k_{i_{2 \to L+1}}$.*

The proof is in App. C, together with the extension to the $s \geq 3$ case (Thm. C.1). It is useful to compare the spectrum in the theorem with the limiting cases of a deep fully-connected network and a shallow CNN. In the former case, the spectrum consists only of the $L$-th sector with $p_L = 1$—the global sector. The eigenvalues decay as $\|\boldsymbol{k}\|^{-2\nu - p}$, with $\nu$ depending ultimately on the nonanalyticity of the network activation function (see Bietti & Bach (2021) or App. C) and $p = d_{\text{eff}}(L)$

the effective dimensionality of the input. As a result, all eigenfunctions with the same $\|\boldsymbol{k}\|$ have the same eigenvalue, even those depending on a subset of the input coordinates. For example, assume that all the components of $\boldsymbol{k}$ are zero but $k_1$, i.e. the eigenfunction depends only on the first 2-dimensional patch: the eigenvalue is $O(k_1^{-2\nu-p})$. By contrast, for a hierarchical kernel, the eigenvalue is $O(k_1^{-2\nu-1})$, much larger than the former as $p > 1$.

In the case of a shallow CNN the spectrum consists only of the first sector, so that each eigenfunction depends only on one of the input patches. In this case only one of the $\boldsymbol{k}$ can be non-zero, say $k_1$, and the eigenvalue is $O(k_1^{-2\nu-1})$. However, from (Favero et al., 2021), a kernel of this kind is only able to approximate functions which depend on one of the input patches or linear combinations of such functions. Instead, for a hierarchical kernel with $p_L = 1$, the eigenfunctions of the $L$-th sector are supported on the full input space. Then, if $\Lambda_{\boldsymbol{k}} > 0$ for all $\boldsymbol{k}$, hierarchical kernels are able to approximate any function on the multisphere, dispensing with the need for fine-tuning the kernel to the structure of the target function.

Overall, given an eigenfunction of a hierarchical kernel, the asymptotic scaling of the corresponding eigenvalue depends on the spatial structure of the eigenfunction support. More specifically, the effective dimensionality of the smallest meta-patch which contains all the variables that the eigenfunction depends on. In simple terms, the decay of an eigenvalue with $k$ is slower if the associated eigenfunction depends on a few adjacent patches—but not if the patches are far apart! This is a property of hierarchical architectures which use nonlinear activation functions at all layers. Such a feature disappears if all hidden layers apart from the first have polynomial (Bietti, 2022) or infinitely smooth (Azevedo & Menegatto, 2015; Scetbon & Harchaoui, 2021) activation functions or if the kernels are assumed to factorise over patches as in Geifman et al. (2022).

## 4 GENERALISATION PROPERTIES AND ADAPTIVITY TO SPATIAL STRUCTURE

In this section, we study the implications of the peculiar spectra of hierarchical NTKs and RFKs on the generalisation properties of and prove a form of adaptivity to the spatial structure of the target function. We follow the classical analysis of Caponnetto & De Vito (2007) for kernel ridge regression (see Bach (2021); Bietti (2022) for a modern treatment) and employ a spectral bias ansatz for the ridgeless limit (Bordelon et al., 2020; Spigler et al., 2020).

**Theory of kernel ridge regression and source-capacity conditions.** Given a set of $n$ training points $\{(\boldsymbol{x}_\mu, y_\mu)\}_{\mu=1}^n \overset{\text{i.i.d.}}{\sim} p(\boldsymbol{x}, y)$ for some probability density function $p(\boldsymbol{x}, y)$ and a regularisation parameter $\lambda > 0$, the kernel ridge regression estimate of the functional relation between $\boldsymbol{x}$'s and $y$'s, or *predictor*, is

$$f_\lambda^n(\boldsymbol{x}) = \underset{f \in \mathcal{H}}{\operatorname{argmin}} \left\{ \frac{1}{n} \sum_{\mu=1}^n (f(\boldsymbol{x}_\mu) - y_\mu)^2 + \lambda \|f\|_{\mathcal{H}} \right\}, \tag{7}$$

where $\mathcal{H}$ is the Reproducing Kernel Hilbert Space (RKHS) of a (hierarchical) kernel $\mathcal{K}$. If $f(\boldsymbol{x})$ denotes the model from which the kernel was obtained via Eq. 4, the space $\mathcal{H}$ is contained in the span of the network features $\{\partial_\theta f(\boldsymbol{x})\}_\theta$ in the infinite-width limit. Alternatively, $\mathcal{H}$ can be defined via the kernel's eigenvalues $\Lambda_{\boldsymbol{k}}$ and eigenfunctions $Y_{\boldsymbol{k}}$: denoting with $f_{\boldsymbol{k}}$ the projections of a function $f$ onto the kernel eigenfunctions, then $f$ belongs to $\mathcal{H}$ if it belongs to the span of the eigenfunctions and

$$\|f\|_{\mathcal{H}}^2 = \sum_{\boldsymbol{k} \geq \boldsymbol{0}} (\Lambda_{\boldsymbol{k}})^{-1} |f_{\boldsymbol{k}}|^2 < +\infty. \tag{8}$$

The performance of the kernel is measured by the generalisation error and its expectation over training sets of fixed size $n$ (denoted with $\mathbb{E}_n$)

$$\epsilon(f_\lambda^n) = \int d\boldsymbol{x} dy\, p(\boldsymbol{x}, y) \left(f_\lambda^n(\boldsymbol{x}) - y\right)^2, \quad \bar{\epsilon}(\lambda, n) = \mathbb{E}_n \left[\epsilon(f_\lambda^n)\right], \tag{9}$$

or the *excess* generalisation error, obtained by subtracting from $\bar{\epsilon}(\lambda, n)$ the error of the optimal predictor $f^*(\boldsymbol{x}) = \int dy\, p(\boldsymbol{x}, y) y$. The decay of the error with $n$ can be controlled via two exponents, depending on the details of the kernel and the target function. Specifically, if $\alpha \geq 1$ and $r \geq 1 - 1/\alpha$

satisfy the following conditions,

$$\text{capacity: } \text{Tr}\left(\mathcal{T}_{\mathcal{K}}^{1/\alpha}\right) = \sum_{\boldsymbol{k}\geq 0}(\Lambda_{\boldsymbol{k}})^{1/\alpha} < +\infty,$$

$$\text{source: } \left\|T_{\mathcal{K}}^{\frac{1-r}{2}}f^*\right\|_{\mathcal{H}}^2 = \sum_{\boldsymbol{k}\geq 0}(\Lambda_{\boldsymbol{k}})^{-r}|f_{\boldsymbol{k}}^*|^2 < +\infty, \tag{10}$$

then, by choosing a $n$-dependent regularisation parameter $\lambda_n \sim n^{-\alpha/(\alpha r+1)}$, one gets the following bound on generalisation (Caponnetto & De Vito, 2007):

$$\bar{\epsilon}(\lambda_n, n) - \epsilon(f^*) \leq \mathcal{C}'n^{-\frac{\alpha r}{\alpha r+1}}. \tag{11}$$

**Spectral bias ansatz for ridgeless regression.** The bound above is actually tight in the noisy setting, for instance when having labels $y_\mu = f^*(\boldsymbol{x}_\mu) + \xi_\mu$ with $\xi_\mu$ Gaussian. In a noiseless problem where $y_\mu = f^*(\boldsymbol{x}_\mu)$ one expects to find the best performances in the ridgeless limit $\lambda \to 0$, so that the rate of Eq. 11 is only an upper bound. In the ridgeless case—where the correspondence between kernel methods and infinitely-wide neural networks actually holds—there are unfortunately no rigorous results for the decay of the generalisation error. Therefore, we provide a heuristic derivation of the error decay based on a spectral bias ansatz. Consider the projections of the target function $f^*$ on the eigenfunctions of the student kernel $Y_{\boldsymbol{k}}$ $(f_{\boldsymbol{k}}^*)$ [3] and assume that kernel methods learn only the $n$ projections corresponding to the highest eigenvalues. Then, if the decay of $f_{\boldsymbol{k}}^*$ with $\boldsymbol{k}$ is sufficiently slow, one has (recall that both $\lambda$ and $\epsilon(f^*)$ vanish in this setting)

$$\bar{\epsilon}(n) \sim \sum_{\boldsymbol{k} \text{ s.t. } \Lambda_{\boldsymbol{k}} < \Lambda(n)}|f_{\boldsymbol{k}}^*|^2, \tag{12}$$

with $\Lambda(n)$ the value of the $n$-th largest eigenvalue of the kernel. This result can be derived using the replica method of statistical physics (see Canatar et al. (2021); Loureiro et al. (2021); Tomasini et al. (2022) and App. E) or by assuming that input points lie on a lattice (Spigler et al., 2020).

These two approaches rely on the very same features of the problem, namely the asymptotic decay of $\Lambda_{\boldsymbol{k}}$ and $|f_{\boldsymbol{k}}^*|^2$—see also Cui et al. (2021). For instance, the capacity condition depends only on the kernel spectrum: $\alpha \geq 1$ since $\text{Tr}(\mathcal{T}_K)$ is finite (Schölkopf et al., 2002); the specific value is determined by the decay of the ordered eigenvalues with their rank, which in turn depends on the scaling of $\Lambda_{\boldsymbol{k}}$ with $\boldsymbol{k}$. Similarly, the power-law decay of the ordered eigenvalues with the rank determines the scaling of the $n$-th largest eigenvalue, $\Lambda(n) \sim n^{-\alpha}$. The source condition characterises the regularity of the target function relative to the kernel and depends explicitly of the decay of $|f_{\boldsymbol{k}}^*|^2$ with $\boldsymbol{k}$, as does the right-hand side of Eq. 12. This condition was used by Bach (2021) to prove that kernel methods are adaptive to the smoothness of the target function: the projections of smoother targets on the eigenfunctions display a faster decay with $\boldsymbol{k}$, thus allowing to choose a larger $r$ and leading to better generalisation performances. The following corollary of Thm. 3.1 (proof and extension to $s_1 \geq 3$ presented in App. D, Cor. D.1) shows that, since the spectrum can be partitioned as in Thm. 3.1, hierarchical kernels display adaptivity to targets which depend only on a subset of the input variables. Specific examples of bounds are considered explicitly in Sec. 5.

**Corollary 4.1 (Adaptivity to spatial structure)** *Let $T_{\mathcal{K}}$ be the integral operator of the kernel of a hierarchical deep CNN as in Thm. 3.1 with $s = 2$. Then:* i) *the capacity exponent $\alpha$ is controlled by the largest sector of the spectrum, i.e.*

$$\text{Tr}\left(\mathcal{T}_{\mathcal{K}}^{1/\alpha}\right) < +\infty \Leftrightarrow \alpha < 1 + 2\nu/d_{\text{eff}}(L); \tag{13}$$

ii) *the source exponent $r$ is controlled by the structure of the target function $f^*$, i.e., if there is $l \leq L$ such that $f^*$ depends only on some meta-patch $\boldsymbol{x}_{i_{l+1}\to L+1}$, then only the first $l$ sectors of the spectrum contribute to the source condition,*

$$\left\|T_{\mathcal{K}}^{\frac{1-r}{2}}f^*\right\|_{\mathcal{H}}^2 = \sum_{l'=1}^{l}\sum_{i_{l'+1\to L+1}}\sum_{\boldsymbol{k}_{i_{l'+1\to L+1}}}\left(\Lambda_{\boldsymbol{k}_{i_{l'+1\to L+1}}}^{(l')}\right)^{-r}\left|f_{\boldsymbol{k}_{i_{l'+1\to L+1}}}^*\right|^2. \tag{14}$$

*The same holds if $f^*$ is a linear combination of such functions. As a result, when $d_{\text{eff}}(L)$ is large and $\alpha \to 1$, the decay of the error is controlled by the effective dimensionality of the target $d_{\text{eff}}(l)$.*

---

[3]We are again limiting the presentation to the case $s = 2$ but the extension to the general case is immediate.

## 5 EXAMPLES AND EXPERIMENTS

**Source-capacity bound for functions of controlled smoothness and $d_{\text{eff}}$.** Consider a target function $f^*$ which only depends on the meta-patch $\boldsymbol{x}_{i_{l+1} \to L+1}$ as in Cor. 4.1. Combining the source condition (Eq. 14) with the asymptotic scaling of eigenvalues (Eq. 6), we get

$$\left\| T_{\mathcal{K}}^{\frac{1-r}{2}} f^* \right\|_{\mathcal{H}}^2 < +\infty \;\Leftrightarrow\; \sum_{\boldsymbol{k}} \|\boldsymbol{k}\|^{r(2\nu + d_{\text{eff}}(l))} |f_{\boldsymbol{k}}^*|^2 < +\infty, \tag{15}$$

where $\nu = 1/2$ (3/2) for the NTK (RFK) and $\boldsymbol{k}$ denotes the meta-patch $\boldsymbol{k}_{i_{l+1} \to L+1}$ without the subscript to ease notation. Since the eigenvalues depend on the norm of $\boldsymbol{k}$, Eq. 15 is equivalent to a finite-norm condition for all the derivatives of $f^*$ up to order $m < r\,(2\nu + d_{\text{eff}}(l))/2$, $\|\Delta^{m/2} f^*\|^2 = \sum_{\boldsymbol{k}} \|\boldsymbol{k}\|^{2m} |f_{\boldsymbol{k}}^*|^2 < +\infty$ with $\Delta$ denoting the Laplace operator. As a result, if $f^*$ has derivatives of finite norm up to the $m$-th, then the source exponent can be tuned to $r = 2m/(2\nu + d_{\text{eff}}(l))$, inversely proportional to the effective dimensionality of $f^*$. Since the exponent on the right-hand side of Eq. 11 is an increasing function of $r$, the smaller the effective dimensionality of $f^*$ the faster the decay of the error—hence hierarchical kernels are adaptive to the spatial structure of $f^*$. In particular, from Eq. 11 with $\alpha = 1 + 2\nu/d_{\text{eff}}(L)$,

$$\bar{\epsilon}(n) \leq \mathcal{C}' n^{-\beta} \text{ with } \beta = \frac{2m\,(2\nu + d_{\text{eff}}(L))}{2m\,(2\nu + d_{\text{eff}}(L)) + (2\nu + d_{\text{eff}}(l))\,d_{\text{eff}}(L)}. \tag{16}$$

For instance, if $p_L = 1$ then $d_{\text{eff}}(L) = p = d/2$ (the number of 2-dim. patches). Even when $p \gg 1$, if $f^*$ depends only on a finite-dimensional meta-patch (or is a sum of such functions) the exponent $\beta$ converges to the finite value $2m/(2(m + \nu) + d_{\text{eff}}(l))$. In stark contrast, using a fully-connected kernel to learn the same target results in $r = 2m/(2\nu + p)$, thus $\beta = 2m/(2m + p)$—vanishing as $1/p$ when $p \gg 1$, thus cursed by dimensionality.

**Rates from spectral bias ansatz.** The same picture emerges when estimating the actual decay of the error from Eq. 12. $\Lambda(n) \sim n^{-\alpha}$, whereas $\sum_{\boldsymbol{k}} \|\boldsymbol{k}\|^{2m} |f_{\boldsymbol{k}}^*|^2 < +\infty$ implies $|f_{\boldsymbol{k}}^*|^2 \lesssim \|\boldsymbol{k}\|^{-2m - d_{\text{eff}}(l)}$ for a target supported on a $d_{\text{eff}}(l)$-dimensional meta-patch. Plugging such decays in Eq. 12 we obtain (details in Subsec. F.1)

$$\bar{\epsilon}(n) \sim n^{-\beta} \text{ with } \beta = \frac{2m}{2\nu + d_{\text{eff}}(l)} \frac{2\nu + d_{\text{eff}}(L)}{d_{\text{eff}}(L)}. \tag{17}$$

Again, with $p_L = 1$ and $d_{\text{eff}}(L) = p$, the exponent remains finite for $p \gg 1$. Notice that we recover the results of Favero et al. (2021) by using a shallow local kernel if the target is supported on $s$-dimensional patches. These results show that hierarchical kernels play significantly better with the approximation-estimation trade-off than shallow local kernels, as they are able to approximate global functions of the input while not being cursed when the target function has a local structure.

**Numerical experiments.** We test our predictions by training a hierarchical kernel (*student*) on a random Gaussian function with zero mean and covariance given by another hierarchical kernel (*teacher*). A learning problem is fully specified by the depths, sets of filter sizes, and smoothness exponents $\nu$ of teacher and student kernels. In particular, the depth and the set of filter sizes of the teacher kernel control the effective dimension of the target function. Fig. 2 shows the learning curves (solid lines) together with the predictions from Eq. 17 (dashed lines), confirming the picture emerging from our calculations. Panel (a) of Fig. 2 shows a depth-four student learning depth-two, depth-three, and depth-four teachers. This student is not cursed in the first two cases and is cursed in the third one, which corresponds to a global target function. Panel (b) illustrates the curse of dimensionality with the effective input dimension $d_{\text{eff}}(L)$ by comparing the learning curves of depth-three students learning global target functions with an increasing number of variables. All our simulations are in excellent agreement with the predictions of Eq. 17. The bounds coming from Eq. 16 would display a slightly slower decay, as sketched in Fig. 1, right panel. All the details of numerical experiments are reported in App. G, together with a comparison between the ridgeless and optimally-regularised cases (Fig. S3) and additional results for: $s_1 \geq 3$ (Fig. S2); kernels with overlapping patches (Fig. S5); different input spaces (Fig. S4) and the CIFAR-10 dataset (Fig. S6).

Notice that when the teacher kernel is a hierarchical RFK, the target is equivalent to the output of a randomly-initialised, infinitely-wide CNN Novak et al. (2019). Although this target is highly structured, it leads to the same rate that we would have obtained for a global non-hierarchical target:

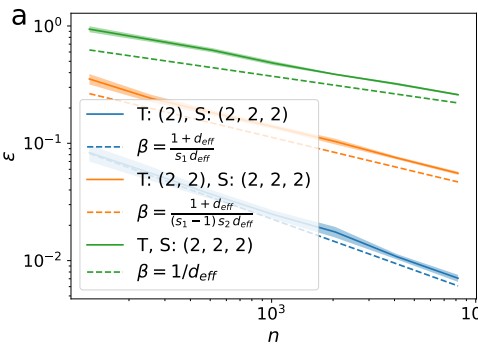 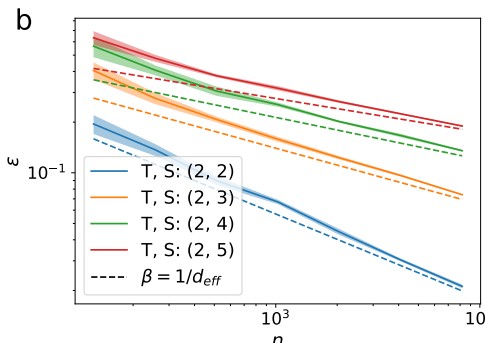

Figure 2: Learning curves for deep convolutional NTKs in a teacher-student setting. **a.** Depth-four student learning depth-two, depth-three, and depth-four teachers. **b.** Depth-three models cursed by the effective input dimensionality $d_{\text{eff}}(L)$. The numbers inside brackets are the sequence of filter sizes of the kernels. Solid lines are the results of experiments averaged over 16 realisations with the shaded areas representing the empirical standard deviations. The predicted asymptotic scaling $\epsilon \sim n^{-\beta}$ are reported as dashed lines. Details on the numerical experiments are reported in App. G.

**Lemma 5.1 (Curse of dimensionality for hierarchical targets)** *The problem of regression of the output of a randomly-initialised and infinitely-wide hierarchical network suffers from the curse of dimensionality, in the sense that no methods using $n$ examples can achieve a generalisation error decaying faster than $n^{-\beta}$ with $\beta = 3/d_{\text{eff}}(L)$.*

This lemma is a simple consequence of the equivalence of the predictors of kernel ridgeless regression and Bayesian inference, which implies that the rate achieved when learning a Gaussian random function with a kernel matching the covariance kernel of the function is Bayes-optimal (Kanagawa et al., 2018). The optimal rate $n^{-3/d_{\text{eff}}(L)}$ comes from Eq. 17 with $l = L$ and $m = \nu = 3/2$. We conclude that, despite their intrinsically hierarchical structure, these targets cannot be good models of learnable tasks.

## 6 CONCLUSIONS AND OUTLOOK

We have proved that deep CNNs can adapt to the spatial scale of the target function, thus beating the curse of dimensionality if the target depends only on local groups of variables. Yet, if considered as 'teachers', they generate functions that cannot be learnt efficiently in high dimensions, even in the Bayes-optimal setting where the student is matched to the teacher. Thus, the architectures we considered are not good models of the hierarchical structure of real data which are efficiently learnable.

Enforcing a stronger notion of compositionality is an interesting endeavour for the future. Following Poggio et al. (2017), one may consider a much smaller family of functions of the form

$$f_{i_{l+1} \to L+1}(\boldsymbol{x}_{i_{l+1} \to L+1}) = f_{i_{l+1} \to L+1}(f_{i_{l=1} \to L+1}(\boldsymbol{x}_{i_{l=1} \to L+1}), \ldots, f_{i_{l=s_l} \to L+1}(\boldsymbol{x}_{i_{l=s_l} \to L+1})). \quad (18)$$

From an information theory viewpoint, Schmidt-Hieber (2020); Finocchio & Schmidt-Hieber (2021) showed that it is possible to learn such functions efficiently. However, these arguments do not provide guarantees for any practical algorithm such as stochastic gradient descent. Moreover, preliminary results (not shown) assuming that the functions $f$'s are random Gaussian functions suggest that these tasks are not learnable efficiently by a hierarchical CNN in the kernel regime—see also Giordano et al. (2022). It is unclear whether this remains true when the networks closely resemble the structure of Eq. 18 as in Poggio et al. (2017), or when the networks are trained in a regime where features can be learnt from data. Recently, for instance, Ingrosso & Goldt (2022) have observed that under certain conditions locality can be learnt from scratch. It is not clear whether compositionality can also be learnt, beyond some very stylised settings (Abbe et al., 2022).

Finally, another direction to explore is the stability of the task toward smooth transformations or diffeomorphisms. This form of stability has been proposed as a key element to understanding how the curse of dimensionality is beaten for image datasets (Bruna & Mallat, 2013; Petrini et al., 2021). Such a property can be enforced with pooling operations (Bietti & Mairal, 2019; Bietti et al., 2021); therefore diagonalising the NTK in this case as well would be of high interest.

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

SUPPLEMENTARY MATERIAL

## A  HARMONIC ANALYSIS ON THE SPHERE

This appendix collects some introductory background on spherical harmonics and dot-product kernels on the sphere Smola et al. (2000). See Efthimiou & Frye (2014); Atkinson & Han (2012); Bach (2017) for a complete description. Spherical harmonics are homogeneous polynomials on the sphere $\mathbb{S}^{s-1} = \{\boldsymbol{x} \in \mathbb{R}^s \mid \|\boldsymbol{x}\| = 1\}$, with $\|\cdot\|$ denoting the L2 norm. Given the polynomial degree $k \in \mathbb{N}$, there are $\mathcal{N}_{k,s}$ linearly independent spherical harmonics of degree $k$ on $\mathbb{S}^{s-1}$, with

$$\mathcal{N}_{k,s} = \frac{2k + s - 2}{k}\binom{s + k - 3}{k - 1}, \quad \begin{cases} \mathcal{N}_{0,d} = 1 \quad \forall d, \\ \mathcal{N}_{k,d} \sim k^{d-2} \quad \text{for } k \gg 1. \end{cases} \tag{S1}$$

Thus, we can introduce a set of $\mathcal{N}_{k,s}$ spherical harmonics $Y_{k,\ell}$ for each $k$, with $\ell$ ranging in $1, \ldots, \mathcal{N}_{k,s}$, which are orthonormal with respect to the uniform measure on the sphere $d\tau(\boldsymbol{x})$,

$$\langle Y_{k,\ell}, Y_{k,\ell'} \rangle_{\mathbb{S}^{s-1}} := \int_{\mathbb{S}^{s-1}} d\tau(\boldsymbol{x}) \, Y_{k,\ell}(\boldsymbol{x}) Y_{k,\ell'}(\boldsymbol{x}) = \delta_{\ell,\ell'}. \tag{S2}$$

Because of the orthogonality of homogeneous polynomials with a different degree, the set $\{Y_{k,\ell}\}_{k,\ell}$ is a complete orthonormal basis for the space of square-integrable functions on the $s$-dimensional unit sphere. Furthermore, spherical harmonics are eigenfunctions of the Laplace-Beltrami operator $\Delta$, which is nothing but the restriction of the standard Laplace operator to $\mathbb{S}^{s-1}$.

$$\Delta Y_{k,\ell} = -k(k+s-2)Y_{k,\ell}. \tag{S3}$$

The Laplace-Beltrami operator $\Delta$ can also be used to characterise the differentiability of functions $f$ on the sphere via the L2 norm of some power of $\Delta$ applied to $f$.

By fixing a direction $\boldsymbol{y}$ in $\mathbb{S}^{d-1}$ one can select, for each $k$, the only spherical harmonic of degree $k$ which is invariant for rotations that leave $\boldsymbol{y}$ unchanged. This particular spherical harmonic is, in fact, a function of $\boldsymbol{x} \cdot \boldsymbol{y}$ and is called the Legendre polynomial of degree $k$, $P_{k,s}(\boldsymbol{x} \cdot \boldsymbol{y})$ (also referred to as Gegenbauer polynomial). Legendre polynomials can be written as a combination of the orthonormal spherical harmonics $Y_{k,\ell}$ via the addition formula (Atkinson & Han, 2012, Thm. 2.9),

$$P_{k,s}(\boldsymbol{x} \cdot \boldsymbol{y}) = \frac{1}{\mathcal{N}_{k,s}} \sum_{\ell=1}^{\mathcal{N}_{k,s}} Y_{k,\ell}(\boldsymbol{x}) Y_{k,\ell}(\boldsymbol{y}). \tag{S4}$$

Alternatively, $P_{k,s}$ is given explicitly as a function of $t = \boldsymbol{x} \cdot \boldsymbol{y} \in [-1, +1]$ via the Rodrigues formula (Atkinson & Han, 2012, Thm. 2.23),

$$P_{k,s}(t) = \left(-\frac{1}{2}\right)^k \frac{\Gamma\left(\frac{s-1}{2}\right)}{\Gamma\left(k+\frac{s-1}{2}\right)} \left(1-t^2\right)^{\frac{3-s}{2}} \frac{d^k}{dt^k} \left(1-t^2\right)^{k+\frac{s-3}{2}}. \tag{S5}$$

Legendre polynomials are orthogonal on $[-1, +1]$ with respect to the measure with density $(1-t^2)^{(s-3)/2}$, which is the probability density function of the scalar product between two points on $\mathbb{S}^{s-1}$.

$$\int_{-1}^{+1} dt \left(1-t^2\right)^{\frac{s-3}{2}} P_{k,s}(t) P_{k',s}(t) = \frac{|\mathbb{S}^{s-1}|}{|\mathbb{S}^{s-2}|} \frac{\delta_{k,k'}}{\mathcal{N}_{k,s}}, \tag{S6}$$

with $|\mathbb{S}^{s-1}|$ denoting the surface area of the $s$-dimensional unit sphere.

To sum up, given $\boldsymbol{x}, \boldsymbol{y} \in \mathbb{S}^{s-1}$, functions of $\boldsymbol{x}$ or $\boldsymbol{y}$ can be expressed as a sum of projections on the orthonormal spherical harmonics $\{Y_{k,\ell}\}_{k,\ell}$, whereas functions of $\boldsymbol{x} \cdot \boldsymbol{y}$ can be expressed as a sum of projections on the Legendre polynomials $\{P_{k,s}(\boldsymbol{x} \cdot \boldsymbol{y})\}_k$. The relationship between the two expansions is elucidated in the Funk-Hecke formula (Atkinson & Han, 2012, Thm. 2.22),

$$\int_{\mathbb{S}^{s-1}} d\tau(\boldsymbol{y}) f(\boldsymbol{x} \cdot \boldsymbol{y}) Y_{k,\ell}(\boldsymbol{y}) = Y_{k,\ell}(\boldsymbol{x}) \frac{|\mathbb{S}^{s-2}|}{|\mathbb{S}^{s-1}|} \int_{-1}^{+1} dt \left(1-t^2\right)^{\frac{s-3}{2}} f(t) P_{k,s}(t). \tag{S7}$$

If the function $f$ has continuous derivatives up to the $k$-th order in $[-1, +1]$, then one can plug Rodrigues' formula in the right-hand side of Funk-Hecke formula and get, after $k$ integrations by parts,

$$\int_{\mathbb{S}^{s-1}} d\tau(\boldsymbol{y}) f(\boldsymbol{x} \cdot \boldsymbol{y}) Y_{k,\ell}(\boldsymbol{y}) = Y_{k,\ell}(\boldsymbol{x}) \frac{|\mathbb{S}^{s-2}|}{|\mathbb{S}^{s-1}|} \frac{\Gamma\left(\frac{s-1}{2}\right)}{2^k \Gamma\left(k+\frac{s-1}{2}\right)} \int_{-1}^{+1} dt \, f^{(k)}(t) \left(1-t^2\right)^{k+\frac{s-3}{2}}, \tag{S8}$$

with $f^{(k)}(t)$ denoting the $k$-th order derivative of $f$ in $t$. This trick also applies to functions which are not $k$ times differentiable at $\pm 1$, provided the boundary terms due to integration by parts vanish.

## A.1 DOT-PRODUCT KERNELS ON THE SPHERE

Dot-product kernels are kernels which depend on the two inputs $\boldsymbol{x}$ and $\boldsymbol{y}$ via their scalar product $\boldsymbol{x} \cdot \boldsymbol{y}$. When the inputs lie on the unit sphere $\mathbb{S}^{s-1}$, one can use the machinery introduced in the previous section to arrive immediately at the Mercer's decomposition of the kernel Smola et al.

(2000).

$$\mathcal{K}(\boldsymbol{x} \cdot \boldsymbol{y}) = \sum_{k \geq 0} \left( \mathcal{N}_{k,s} \frac{|\mathbb{S}^{s-2}|}{|\mathbb{S}^{s-1}|} \int_{-1}^{+1} dt \left( 1 - t^2 \right)^{\frac{s-3}{2}} \mathcal{K}(t) P_{k,s}(t) \right) P_{k,s}(\boldsymbol{x} \cdot \boldsymbol{y})$$

$$= \sum_{k \geq 0} \left( \frac{|\mathbb{S}^{s-2}|}{|\mathbb{S}^{s-1}|} \int_{-1}^{+1} dt \left( 1 - t^2 \right)^{\frac{s-3}{2}} \mathcal{K}(t) P_{k,s}(t) \right) \sum_{\ell=1}^{\mathcal{N}_{k,s}} Y_{k,\ell}(\boldsymbol{x}) Y_{k,\ell}(\boldsymbol{y}) \qquad \text{(S9)}$$

$$:= \sum_{k \geq 0} \Lambda_k \sum_{\ell=1}^{\mathcal{N}_{k,s}} Y_{k,\ell}(\boldsymbol{x}) Y_{k,\ell}(\boldsymbol{y}).$$

In the first line we have just decomposed $\mathcal{K}$ into projections onto the Legendre polynomials, the second line follows immediately from the addition formula, and the third is just a definition of the eigenvalues $\Lambda_k$. Notice that the eigenfunctions of the kernel are orthonormal spherical harmonics and the eigenvalues are degenerate with respect to the index $\ell$. The Reproducing Kernel Hilbert Space (RKHS) of $\mathcal{K}$ can be characterised as follows,

$$\mathcal{H} = \left\{ f : \mathbb{S}^{s-1} \to \mathbb{R} \text{ s. t. } \|f\|_{\mathcal{H}} := \sum_{k \geq 0, \Lambda_k \neq 0} \sum_{\ell=1}^{\mathcal{N}_{k,s}} \frac{\langle f, Y_{k,l} \rangle^2_{\mathbb{S}^{s-1}}}{\Lambda_k} < +\infty \right\}. \qquad \text{(S10)}$$

### A.2 MULTI-DOT-PRODUCT KERNELS ON THE MULTI-SPHERE

Mercer's decomposition of dot-product kernels extends naturally to the case considered in this paper, where the input space is the Cartesian product of $p$ $s$-dimensional unit sphere,

$$\mathsf{M}^p \mathbb{S}^{s-1} = \left\{ \boldsymbol{x} = (\boldsymbol{x}_1, \ldots, \boldsymbol{x}_p) \, \big| \, \boldsymbol{x}_i \in \mathbb{S}^{s-1} \, \forall \, i = 1, \ldots, p \right\} = \bigtimes_{i=1}^{p} \mathbb{S}^{s-1} \qquad \text{(S11)}$$

which we refer to as the *multi-sphere* following the notation of Geifman et al. (2022). After defining a scalar product between functions on $\mathsf{M}^p \mathbb{S}^{s-1}$ by direct extension of Eq. S2, one can immediately find a set of orthonormal polynomials by taking products of spherical harmonics. With the multi-index notation $\boldsymbol{k} = (k_1, \ldots, k_p)$, $\boldsymbol{\ell} = (\ell_1, \ldots, \ell_p)$, for all $\boldsymbol{x} \in \mathsf{M}^p \mathbb{S}^{s-1}$

$$\tilde{Y}_{\boldsymbol{k},\boldsymbol{\ell}}(\boldsymbol{x}) = \prod_{i=1}^{p} Y_{k_i,\ell_i}(\boldsymbol{x}_i), \text{ with } k_i \geq 0, \ell_i = 1, \ldots, \mathcal{N}_{k_i,s} = \frac{2k_i + s - 2}{k_i} \binom{s + k_i - 3}{k_i - 1}. \qquad \text{(S12)}$$

These product spherical harmonics $\tilde{Y}_{\boldsymbol{k},\boldsymbol{\ell}}(\boldsymbol{x})$ span the space of square-integrable functions on $\mathsf{M}^p \mathbb{S}^{s-1}$. Furthermore, as each spherical harmonic is an eigenfunction of the Laplace-Beltrami operator, $\tilde{Y}_{\boldsymbol{k},\boldsymbol{\ell}}$ is an eigenfunction of the sum of Laplace-Beltrami operators on the $p$ unit spheres,

$$\Delta_{p,s} \tilde{Y}_{\boldsymbol{k},\boldsymbol{\ell}} := \left( \sum_{i=1}^{p} \Delta_i \right) \prod_{i=1}^{p} Y_{k_i,\ell_i} = \left( \sum_{i=1}^{p} ((-k_i)(k_i + s - 2)) \right) \tilde{Y}_{\boldsymbol{k},\boldsymbol{\ell}}. \qquad \text{(S13)}$$

We can thus characterise the differentiability of functions of the multi-sphere $\mathcal{X}_{s,p}$ via finiteness in L2 norm of some power of $\Delta_{p,s}$.

Similarly, we can consider products of Legendre polynomials to obtain a set of orthogonal polynomials on $[-1, 1]^p$ (see Geifman et al. (2022), appendix A). Then, any function $f$ on $\mathsf{M}^p \mathbb{S}^{s-1} \times \mathsf{M}^p \mathbb{S}^{s-1}$ which depends only on the $p$ scalar products between patches,

$$f(\boldsymbol{x}, \boldsymbol{y}) = g(\boldsymbol{x}_1 \cdot \boldsymbol{y}_1, \ldots, \boldsymbol{x}_p \cdot \boldsymbol{y}_p), \qquad \text{(S14)}$$

can be written as a sum of projections on products of Legendre polynomials

$$\tilde{P}_{\boldsymbol{k},s}(\boldsymbol{t}) := \prod_{i=1}^{p} P_{k_i,s}(t_i). \qquad \text{(S15)}$$

Following Geifman et al. (2022), we call such functions *multi-dot-product* kernels. When fixing one of the two arguments of $f$ (say $\boldsymbol{x}$), $f$ becomes a function on $\mathsf{M}^p \mathbb{S}^{s-1} \times \mathsf{M}^p \mathbb{S}^{s-1}$ and can be written

as a sum of projections on the $\tilde{Y}_{\boldsymbol{k},\boldsymbol{\ell}}$'s. The two expansions are related by the following generalised Funk-Hecke formula,

$$
\left( \prod_{i=1}^{p} \int_{\mathbb{S}^{s-1}} d\tau(\boldsymbol{y}_i) \right) g(\boldsymbol{x}_1 \cdot \boldsymbol{y}_1, \ldots, \boldsymbol{x}_p \cdot \boldsymbol{y}_p) \tilde{Y}_{\boldsymbol{k},\boldsymbol{\ell}}(\boldsymbol{y}) =
$$
$$
\tilde{Y}_{\boldsymbol{k},\boldsymbol{\ell}}(\boldsymbol{y}) \left( \frac{|\mathbb{S}^{s-2}|}{|\mathbb{S}^{s-1}|} \right)^p \left( \prod_{i=1}^{p} \int_{-1}^{+1} dt_i \left(1 - t_i^2\right)^{\frac{s-3}{2}} P_{k_i,s}(t_i) \right) g(t_1, \ldots, t_p).
$$
(S16)

Having introduced the product spherical harmonics $\tilde{Y}_{\boldsymbol{k},\boldsymbol{\ell}}$ as basis of $\mathsf{M}^p\mathbb{S}^{s-1}$ and the product Legendre polynomials $\tilde{P}_{\boldsymbol{k},s}(\boldsymbol{t})$ as basis of $[-1,+1]^p$, the Mercer's decomposition of multi-dot-product kernels follows immediately.

$$
\mathcal{K}\left(\{\boldsymbol{x}_i \cdot \boldsymbol{y}_i\}_i\right) = \sum_{\boldsymbol{k} \geq 0} \left( \prod_{i=1}^{p} \mathcal{N}_{k_i,s} \frac{|\mathbb{S}^{s-2}|}{|\mathbb{S}^{s-1}|} \int_{-1}^{+1} dt_i \left(1 - t_i^2\right)^{\frac{s-3}{2}} P_{k_i,s}(t_i) \right) \mathcal{K}\left(\{t_i\}_i\right) P_{\boldsymbol{k},s}\left(\{\boldsymbol{x}_i \cdot \boldsymbol{y}_i\}_i\right)
$$
$$
= \sum_{k \geq 0} \Lambda_k \sum_{\ell=1}^{\mathcal{N}_{k,s}} Y_{k,\ell}(\boldsymbol{x}) Y_{k,\ell}(\boldsymbol{y}).
$$
(S17)

# B   RFK AND NTK OF DEEP CONVOLUTIONAL NETWORKS

This appendix gives the functional forms of the RFK and NTK of hierarchical CNNs. We refer the reader to Arora et al. (2019) for the derivation.

**Definition B.1 (RFK and NTK of hierarchical CNNs)** *Let $\boldsymbol{x}, \boldsymbol{y} \in \mathsf{M}^p\mathbb{S}^{s-1} = \prod_{i=1}^{p}\mathbb{S}^{s-1}$. Denote tuples of the kind $i_l i_{l+1} \ldots i_m$ with $i_{l \to m}$ for $m \geq l$. For $m < l$, $i_{l \to m}$ denotes the empty tuple. For each tuple $i_{2 \to L+1}$, denote with $t_{i_{2 \to L+1}}$ the scalar product between the $s$-dimensional patches of $\boldsymbol{x}$ and $\boldsymbol{y}$ identified by the same tuple, i.e.*

$$
t_{i_{2 \to L+1}} = \boldsymbol{x}_{i_{2 \to L+1}} \cdot \boldsymbol{y}_{i_{2 \to L+1}}
$$
(S18)

*For $1 \leq l \leq L+1$, denote with $\left\{t_{i_{2 \to L+1}}\right\}_{i_{2 \to l}}$ the sequence of $t$'s obtained by letting the indices of the tuple $i_{2 \to l}$ vary in their respective range. Consider a hierarchical CNN with $L$ hidden layers, filter sizes $(s_1, \ldots, s_L)$, $p_L \geq 1$ and all the weights $w_{h,i}^{(1)}, w_{h,h',i}^{(l)}, w_{h,i}^{(L+1)}$ initialised as Gaussian random numbers with zero mean and unit variance.*

**RFK.** *The corresponding RFK (or covariance kernel) is a function $\mathcal{K}_{\mathrm{RFK}}^{(L+1)}$ of the $p_1 = d/s_1$ scalar products $t_{i_L \ldots i_1}$ which can be obtained recursively as follows. With $\kappa_1(t) = \left((\pi - \arccos t)\, t + \sqrt{1 - t^2}\right)/\pi$,*

$$
\mathcal{K}_{\mathrm{RFK}}^{(1)}(t_{i_{2 \to L+1}}) = \kappa_1(t_{i_{2 \to L+1}});
$$
$$
\mathcal{K}_{\mathrm{RFK}}^{(l)}\left(\left\{t_{i_{2 \to L+1}}\right\}_{i_{2 \to l}}\right) = \kappa_1\left(\frac{1}{s_l}\sum_{i_l} \mathcal{K}_{\mathrm{RFK}}^{(l-1)}\left(\left\{t_{i_{2 \to L+1}}\right\}_{i_{2 \to l-1}}\right)\right), \; \forall l \in [2 \mathinner{\ldotp\ldotp} L] \; \text{if } L > 1;
$$
$$
\mathcal{K}_{\mathrm{RFK}}^{(L+1)}\left(\left\{t_{i_{2 \to L+1}}\right\}_{i_{2 \to L+1}}\right) = \frac{1}{p_L}\sum_{i_{L+1}=1}^{p_L} \mathcal{K}_{\mathrm{RFK}}^{(L)}\left(\left\{t_{i_{2 \to L+1}}\right\}_{i_{2 \to L}}\right).
$$
(S19)

**NTK.** *The NTK of the same hierarchical CNN is also a function of the $p_1 = d/s_1$ scalar products $t_{i_L \ldots i_2}$ which can be obtained recursively as follows. With $\kappa_0(t) = (\pi - \arccos t)/\pi$,*

$$\mathcal{K}_{\mathrm{NTK}}^{(1)}\left(t_{i_{2 \to L+1}}\right) = \kappa_1(t_{i_{2 \to L+1}}) + \left(t_{i_{2 \to L+1}}\right)\kappa_0(t_{i_{2 \to L+1}});$$

$$\mathcal{K}_{\mathrm{NTK}}^{(l)}\left(\{t_{i_{2 \to L+1}}\}_{i_{2 \to l}}\right) = \mathcal{K}_{\mathrm{RFK}}^{(l)}(\{t_{i_{2 \to L+1}}\}_{i_{2 \to l}}) + \left(\frac{1}{s_l}\sum_{i_l}\mathcal{K}_{\mathrm{NTK}}^{(l-1)}\left(\{t_{i_{2 \to L+1}}\}_{i_{2 \to l-1}}\right)\right)$$

$$\times \kappa_0\left(\frac{1}{s_l}\sum_{i_l}\mathcal{K}_{\mathrm{RFK}}^{(l-1)}\left(\{t_{i_{2 \to L+1}}\}_{i_{2 \to l-1}}\right)\right), \ \forall l \in [2 \ldots L] \ \text{if } L > 1;$$

$$\mathcal{K}_{\mathrm{NTK}}^{(L+1)}\left(\{t_{i_{2 \to L+1}}\}_{i_{2 \to L+1}}\right) = \frac{1}{p_L}\sum_{i_{L+1}=1}^{p_L}\mathcal{K}_{\mathrm{NTK}}^{(L)}\left(\{t_{i_{2 \to L+1}}\}_{i_{2 \to L}}\right). \tag{S20}$$

## C  Spectra of deep convolutional kernels

In this section we state and prove a generalised version of Thm. 3.1 which includes non-binary patches. Our proof strategy is to relate the asymptotic decay of eigenvalues to the singular behaviour of the kernel, as it is customary in Fourier analysis and was done in Bietti & Bach (2021) for standard dot-product kernel. In Subsec. C.1 we perform the singular expansion of hierarchical kernels, in Subsec. C.2 we use this expansion to prove Thm. 3.1 with $L = 2$ (2 hidden layers) and $s_1 = 2$ (patches on the ring), which we then generalise to general $s_1$ in Subsec. C.3 and to general depth in Subsec. C.4.

**Theorem C.1 (Spectrum of hierarchical kernels)** *Let $T_{\mathcal{K}}$ be the integral operator associated with a $d$-dimensional hierarchical kernel of depth $L + 1$, $L > 1$ and filter sizes $(s_1, \ldots, s_L)$. Eigenvalues and eigenfunctions of $T_{\mathcal{K}}$ can be organised into $L$ sectors associated with the hidden layers of the kernel/network. For each $1 \le l \le L$, the $l$-th sector consists of $(\prod_{l'=1}^{l} s_{l'})$-local eigenfunctions: functions of a single meta-patch $\boldsymbol{x}_{i_{l+1 \to L+1}}$ which cannot be written as linear combinations of functions of smaller meta-patches. The labels $\boldsymbol{k}$ of these eigenfunctions are such that there is a meta-patch $\boldsymbol{k}_{i_{l+1 \to L+1}}$ of $\boldsymbol{k}$ with no vanishing sub-meta-patches and all the $k_i$'s outside of $\boldsymbol{k}_{i_{l+1 \to L+1}}$ are $0$ (because the eigenfunction is constant outside of $\boldsymbol{x}_{i_{l+1 \to L+1}}$). The corresponding eigenvalue is degenerate with respect to the location of the meta-patch: we call it $\Lambda_{\boldsymbol{k}_{i_{l+1 \to i_L+1}}}^{(l)}$. When $\|\boldsymbol{k}_{i_{l+1 \to L+1}}\| \to \infty$, with $k = \|\boldsymbol{k}_{i_{l+1 \to L+1}}\|$,*

   i. *if $s_1 = 2$, then*

$$\Lambda_{\boldsymbol{k}_{i_{l+1 \to L+1}}}^{(l)} = \mathcal{C}_{2,l}\, k^{-2\nu - d_{\mathrm{eff}}(l)} + o\left(k^{-2\nu - d_{\mathrm{eff}}(l)}\right), \tag{S21}$$

   *with $\nu_{\mathrm{NTK}} = 1/2$, $\nu_{\mathrm{RFK}} = 3/2$ and $d_{\mathrm{eff}}$ the effective dimensionality of the meta-patches defined in Eq. 3. $\mathcal{C}_{2,l}$ is a strictly positive constant for $l \ge 2$ whereas for $l = 1$ it can take two distinct strictly positive values depending on the parity of $k_{i_{2 \to L+1}}$.*

   ii. *if $s_1 \ge 3$, then for fixed non-zero angles $\boldsymbol{k}/k$,*

$$\Lambda_{\boldsymbol{k}_{i_{l+1 \to L+1}}}^{(l)} = \mathcal{C}_{s_1,l}\left(\frac{\boldsymbol{k}_{i_{l+1 \to L+1}}}{k}\right) k^{-2\nu - d_{\mathrm{eff}}(l)} + o\left(k^{-2\nu - d_{\mathrm{eff}}(l)}\right), \tag{S22}$$

   *where $\mathcal{C}_{s_1,l}$ is a positive function for $l \ge 2$, whereas for $l = 1$ it is a strictly positive constant which depends on the parity of $k_{i_{2 \to L+1}}$.*

### C.1  Singular expansion of hierarchical kernels

Both the RFK and NTK of ReLU networks, whether deep or shallow, are built by applying the two functions $\kappa_0$ and $\kappa_1$ Cho & Saul (2009) (see also Def. B.1),

$$\kappa_0(t) = \frac{(\pi - \arccos t)}{\pi}, \quad \kappa_1(t) = \frac{(\pi - \arccos t)\, t + \sqrt{1 - t^2}}{\pi}. \tag{S23}$$

The functions $\kappa_0$ and $\kappa_1$ are non-analytic in $t = \pm 1$, with the following singular expansion Bietti & Bach (2021). Near $t = 1$, with $u = 1 - t$

$$\begin{cases} \kappa_0(1-u) = 1 - \dfrac{\sqrt{2}}{\pi} u^{1/2} + O(u^{3/2}), \\[2mm] \kappa_1(1-u) = 1 - u + \dfrac{2\sqrt{2}}{3\pi} u^{3/2} + O(u^{5/2}). \end{cases} \tag{S24}$$

Near $t = -1$, with $u = 1 + t$,

$$\begin{cases} \kappa_0(-1+u) = \dfrac{\sqrt{2}}{\pi} u^{1/2} + O(u^{3/2}), \\[2mm] \kappa_1(-1+u) = \dfrac{2\sqrt{2}}{3\pi} u^{3/2} + O(u^{5/2}). \end{cases} \tag{S25}$$

As a result, hierarchical kernels have a singular expansion when the $t_{i_2 \to L+1}$'s are close to $\pm 1$. In particular, the following expansions are relevant for computing the asymptotic scaling of eigenvalues.

**Proposition C.1 (RFK when $x = y$)** *The RFK of a hierarchical network of depth $L+1$, filter sizes $(s_1, \ldots, s_L)$ and $p_L \geq 1$ has the following singular expansion when all $t_{i_2 \to L+1} \to 1$. With $u_{i_2 \to L+1} = 1 - t_{i_2 \to L+1}$, $c = 2\sqrt{2}/(3\pi)$, and $\prod_{l \in I} s_l := 1$ if $I$ is the empty set,*

$$\mathcal{K}_{\mathrm{RFK}}^{(L+1)} \left( \{ 1 - u_{i_2 \to L+1} \}_{i_2 \to L+1} \right) = 1 - \frac{1}{\left( \displaystyle\prod_{2 \leq l' \leq L} s_{l'} \right) p_L} \sum_{i_2 \to L+1} u_{i_2 \to L+1}$$

$$+ \frac{c}{p_L} \sum_{l'=1}^{L} \frac{1}{\left( \displaystyle\prod_{l' < l'' \leq L} s_{l''} \right)} \sum_{i_{l'+1} \to L+1} \left( \frac{\sum_{i_2 \to l'} u_{i_2 \to L+1}}{\left( \displaystyle\prod_{2 \leq l'' \leq l'} s_{l''} \right)} \right)^{3/2}$$

$$+ O(u_{i_2 \to L+1}^{5/2}) \tag{S26}$$

*Proof.* With $L = 1$ one has (recall that $i_{2 \to 1+1} = i_{2 \to 2}$ reduces to a single index)

$$\mathcal{K}_{\mathrm{RFK}}^{(1)}(1 - u_{i_2}) = 1 - u_{i_2} + c u_{i_2}^{3/2} + O(u_{i_2}^{5/2}) \Rightarrow$$

$$\mathcal{K}_{\mathrm{RFK}}^{(1+1)} \left( \{ 1 - u_{i_2} \}_{i_2} \right) = 1 - \frac{1}{p_1} \sum_{i_2} u_{i_2} + \frac{c}{p_1} \sum_{i_2} u_{i_2}^{3/2} + O(u_{i_2}^{5/2}). \tag{S27}$$

With $L = 2$,

$$\mathcal{K}_{\mathrm{RFK}}^{(2)} \left( \{ 1 - u_{i_2} \}_{i_2} \right) = \kappa_1 \left( 1 - \frac{1}{s_2} \sum_{i_2} u_{i_2, i_3} + \frac{c}{s_2} \sum_{i_2} u_{i_2, i_3}^{3/2} + O(u_{i_2, i_3}^{5/2}) \right)$$

$$= 1 - \frac{1}{s_2} \sum_{i_2} u_{i_2, i_3} + \frac{c}{s_2} \sum_{i_2} u_{i_2, i_3}^{3/2} + c \left( \frac{1}{s_2} \sum_{i_2} u_{i_2, i_3} \right)^{3/2} + O(u_{i_2, i_3}^{5/2}), \tag{S28}$$

therefore

$$\mathcal{K}_{\mathrm{RFK}}^{(2+1)} \left( \{ 1 - u_{i_2, i_3} \}_{i_2, i_3} \right) = 1 - \frac{1}{s_2 p_2} \sum_{i_2, i_3} u_{i_2, i_3} + \frac{c}{p_2} \frac{1}{s_2} \sum_{i_2, i_3} u_{i_2, i_3}^{3/2} + \frac{c}{p_2} \sum_{i_3} \left( \frac{1}{s_2} \sum_{i_2} u_{i_2, i_3} \right)^{3/2}$$

$$+ O(u_{i_2, i_3}^{5/2}). \tag{S29}$$

The proof of the general case follows by induction by applying the function $\kappa_1$ to the singular expansion of the kernel with $L - 1$ hidden layers, then using Eq. S24.

**Proposition C.2 (RFK when $x = -y$)** *The RFK of a hierarchical network of depth $L+1$, filter sizes $(s_1, \ldots, s_L)$ and $p_L \geq 1$ has the following singular expansion when all $t_{i_2 \to L+1} \to -1$. With $u_{i_2 \to L+1} = 1 + t_{i_2 \to L+1}$, $c = 2\sqrt{2}/(3\pi)$ and $\prod_{l \in I} s_l := 1$ if $I$ is the empty set,*

$$\mathcal{K}_{\text{RFK}}^{(L+1)}\left(\{-1 + u_{i_2 \to L+1}\}_{i_2 \to L+1}\right) = b_L + \frac{c_L}{\left(\displaystyle\prod_{2 \leq l' \leq L} s_{l'}\right) p_L} \sum_{i_2 \to L+1} u_{i_2 \to L+1}^{3/2} + O(u_{i_2 \to L+1}^{5/2}),$$

(S30)

*with $b_L = \kappa_1(b_{L-1})$, $b_1 = 0$; and $c_L = c_{L-1}\kappa_1'(b_{L-1})$, $c_1 = c$.*

*Proof.* This can be proved again by induction. For $L = 1$,

$$\mathcal{K}_{\text{RFK}}^{(1)}(-1 + u_{i_2}) = cu_{i_2}^{3/2} + O(u_{i_2}^{5/2}) \Rightarrow$$
$$\mathcal{K}_{\text{RFK}}^{(1+1)}\left(\{-1 + u_{i_2}\}_{i_2}\right) = \frac{c}{p_1} \sum_{i_2} u_{i_2}^{3/2} + O(u_{i_2}^{5/2}).$$

(S31)

Thus, for $L = 2$,

$$\mathcal{K}_{\text{RFK}}^{(2)}\left(\{-1 + u_{i_2, i_3}\}_{i_2}\right) = \kappa_1\left(\frac{c}{s_2} \sum_{i_2} u_{i_2, i_3}^{3/2} + O(u_{i_2, i_3}^{5/2})\right)$$

$$= \kappa_1(0) + \kappa_1'(0)\left(\frac{c}{s_2} \sum_{i_2} u_{i_2, i_3}^{3/2}\right) + O(u_{i_2, i_3}^{5/2}),$$

(S32)

so that

$$\mathcal{K}_{\text{RFK}}^{(2+1)}\left(\{-1 + u_{i_2, i_3}\}_{i_2, i_3}\right) = \kappa_1(0) + \frac{\kappa_1'(0)c}{s_2 p_2} \sum_{i_2, i_3} u_{i_2, i_3}^{3/2} + O(u_{i_2, i_3}^{5/2}).$$

(S33)

The proof is completed by applying the function $\kappa_1$ to the singular expansion of the kernel with $L - 1$ hidden layers.

**Proposition C.3 (NTK when $x = y$)** *The NTK of a hierarchical network of depth $L+1$, filter sizes $(s_1, \ldots, s_L)$ and $p_L \geq 1$ has the following singular expansion when all $t_{i_2 \to L+1} \to 1$. With $u_{i_2 \to L+1} = 1 - t_{i_2 \to L+1}$, $c = \sqrt{2}\pi$, and $\prod_{l \in I} s_l := 1$ if $I$ is the empty set,*

$$\mathcal{K}_{\text{NTK}}^{(L+1)}\left(\{1 - u_{i_2 \to L+1}\}_{i_2 \to L+1}\right) = L + 1 - \frac{c}{p_L} \sum_{l'=1}^{L} \frac{l'}{\left(\displaystyle\prod_{l' < l'' \leq L} s_{l''}\right)}$$

$$\times \sum_{i_{l'+1} \to L+1} \left(\frac{1}{\left(\displaystyle\prod_{2 \leq l'' \leq l'} s_{l''}\right)} \sum_{i_2 \to l'} u_{i_2 \to L+1}\right)^{1/2} + O(u_{i_2 \to L+1}^{3/2})$$

(S34)

**Proposition C.4 (NTK when $x = -y$)** *The NTK of a hierarchical network of depth $L+1$, filter sizes $(s_1, \ldots, s_L)$ and $p_L \geq 1$ has the following singular expansion when all $t_{i_2 \to L+1} \to -1$. With $u_{i_2 \to L+1} = 1 + t_{i_2 \to L+1}$, $c = \sqrt{2}/\pi$ and $\prod_{l \in I} s_l := 1$ if $I$ is the empty set,*

$$\mathcal{K}_{\text{NTK}}^{(L+1)}\left(\{-1 + u_{i_2 \to L+1}\}_{i_2 \to L+1}\right) = a_L + \frac{c_L}{\left(\displaystyle\prod_{2 \leq l' \leq L} s_{l'}\right) p_L} \sum_{i_2 \to L+1} u_{i_2 \to L+1}^{3/2} + O(u_{i_2 \to L+1}^{5/2}),$$

(S35)

*with $a_L = b_L + b_{L-1}\kappa_0(b_{L-1})$, $b_L = \kappa_1(b_{L-1})$, $b_1 = 0$; and $c_L = c_{L-1}\kappa_0(b_{L-1})$, $c_1 = c$. Notice that both $\kappa_1$ and $\kappa_0$ are positive and strictly increasing in $[0, 1]$ and $\kappa_1(1) = \kappa_0(1) = 1$, thus $b_L \in (0, 1)$ and $c_L < c_{L-1}$.*

The proofs of the two propositions above are omitted, as they follow the exact same steps as the previous two proofs.

### C.2 PATCHES ON THE RING

In this section, we prove a restricted version of Thm. 3.1 for the case of 2-dimensional input patches, since the reduction of spherical harmonics to the Fourier basis simplifies the proof significantly. We also consider, for convenience, hierarchical kernels of depth 3 with the filter size of the second hidden layer set to $p = d/2$, the total number of 2-patches of the input. Once this case is understood, extension to arbitrary filter size and arbitrary depth is trivial.

**Theorem C.2 (Spectrum of depth-3 kernels on 2-patches)** *Let $T_{\mathcal{K}}$ be the integral operator associated with a $d$-dimensional hierarchical kernel of depth 3, (2 hidden layers), with filter sizes $(s_1 = 2, s_2)$ and $p_2 = 1$, such that $2s_2 = d$ and $s_2 = p$ (the number of 2-patches). Eigenvalues and eigenfunctions of $T_{\mathcal{K}}$ can be organised into 2 sectors associated with the hidden layers of the kernel/network.*

   i. *The first sector consists of $s_1$-local eigenfunctions, which are functions of a single patch $\boldsymbol{x}_i$ for $i = 1, \ldots, p$. The labels $\boldsymbol{k}, \boldsymbol{\ell}$ of local eigenfunctions are such that all the $k_j$'s with $j \neq i$ are zero (because the eigenfunction is constant outside $\boldsymbol{x}_i$). The corresponding eigenvalue is degenerate with respect to the location of the patch: we call it $\Lambda_{k_i}^{(1)}$. When $k_i \to \infty$,*

$$\Lambda_{k_i}^{(1)} = \mathcal{C}_{2,1} k^{-2\nu-1} + o\left(k^{-2\nu-1}\right), \tag{S36}$$

   *with $\nu_{\mathrm{NTK}} = 1/2$, $\nu_{\mathrm{RFK}} = 3/2$. $\mathcal{C}_{2,l}$ can take two distinct strictly positive values depending on the parity of $k_i$;*

   ii. *The second sector consists of global eigenfunctions, which are functions of the whole input $\boldsymbol{x}$. The labels $\boldsymbol{k}, \boldsymbol{\ell}$ of global eigenfunctions are such that at least two of the $k_i$'s are non-zero. We call the corresponding eigenvalue $\Lambda_{\boldsymbol{k}}^{(2)}$. When $\|\boldsymbol{k}\| \to \infty$, with $k = \|\boldsymbol{k}\|$,*

$$\Lambda_{\boldsymbol{k}}^{(2)} = \mathcal{C}_{2,2} k^{-2\nu-p} + o\left(k^{-2\nu-p}\right), \tag{S37}$$

*Proof.* If we consider binary patches in the first layer, the input space becomes the Cartesian product of two-dimensional unit spheres, i.e. circles, $\mathcal{X} = \prod_{i=1}^{d} \mathbb{S}^1$. Then, each patch $\boldsymbol{x}_i$ corresponds to an angle $\theta_i$ and the spherical harmonics are equivalent to Fourier atoms,

$$Y_0(\theta) = 1, \quad Y_{k,1}(\theta) = e^{ik\theta}, \quad Y_{k,2}(\theta) = e^{-ik\theta}, \quad \forall k \geq 1. \tag{S38}$$

Therefore, solving the eigenvalue problem for a dot-product kernel $\mathcal{K}(\boldsymbol{x} \cdot \boldsymbol{y}) = \mathcal{K}(\cos(\theta_x - \theta_y))$ with $\boldsymbol{x}, \boldsymbol{y} \in \mathbb{S}^1$ reduces to computing its Fourier transform. With $|\mathbb{S}^0| = 2$ and $|\mathbb{S}^1| = 2\pi$,

$$\frac{1}{2\pi} \int_{-\pi}^{\pi} d\theta_x \, \mathcal{K}\left(\cos(\theta_x - \theta_y)\right) e^{\pm ik\theta_x} = \Lambda_k e^{\pm ik\theta_y} \Rightarrow \Lambda_k = \frac{1}{2\pi} \int_{-\pi}^{\pi} d\theta \, \mathcal{K}\left(\cos\theta\right) e^{\pm ik\theta}, \tag{S39}$$

where we denoted with $\theta$ the difference between the two angles. Similarly, for a multi-dot-product kernel, the eigenvalues coincide with the $p$-dimensional Fourier transform of the kernel, where $p$ is the number of patches,

$$\Lambda_{\boldsymbol{k}} = \frac{1}{(2\pi)^p} \int_{-\pi}^{\pi} \left(\prod_{i=1}^{p} d\theta_i \, e^{\pm ik_i\theta_i}\right) \mathcal{K}\left(\{\cos\theta_i\}_{i=1}^{p}\right)$$

$$= \frac{1}{(2\pi)^p} \int_{-\pi}^{\pi} d^p\boldsymbol{\theta} \, e^{\pm i\boldsymbol{k}\cdot\boldsymbol{\theta}} \mathcal{K}\left(\{\cos\theta_i\}_{i=1}^{p}\right), \tag{S40}$$

with $\boldsymbol{k} = (k_1, \ldots, k_p)^\top$ the vector of the patch wavevectors and $\boldsymbol{\theta} = (\theta_1, \ldots, \theta_p)^\top$ the vector of the patch angle differences $\theta_i = \theta_{x,i} - \theta_{y,i}$.

The nonanaliticity of the kernel at $t_i = 1$ for all $i$ moves to $\theta_i = 0$ for all $i$, whereas those in $t_i = -1$ move to $\theta_i = \pi$ and $-\pi$. The corresponding singular expansion is obtained from Eq. S26 after replacing $t_i$ with $\cos{(\theta_i)}$ and expanding $\cos{(\theta_i)}$ as $1 - \theta_i^2/2$, resulting in

$$\mathcal{K}_{\mathrm{RFK}}^{(2)}(\{\cos\theta_i\}_{i=1}^p) = 1 - \frac{1}{2p}\sum_{i=1}^p \theta_i^2 + \frac{1}{3\pi p}\sum_{i=1}^p |\theta_i|^3 + \frac{2\sqrt{2}}{3\pi}\left(\frac{1}{p}\sum_{i=1}^p \frac{\theta_i^2}{2}\right)^{3/2} + \sum_{i=1}^p O(\theta_i^4). \quad \text{(S41)}$$

The first nonanalytic terms are $\frac{1}{3\pi p}\sum_{i=1}^p |\theta_i|^3$ and $\frac{2\sqrt{2}}{3\pi}\left(\frac{1}{p}\sum_{i=1}^p \frac{\theta_i^2}{2}\right)^{3/2}$. After recalling that the Fourier transform of $\|\boldsymbol{\theta}\|^{2\nu}$ with $\boldsymbol{\theta} \in \mathbb{R}^p$ decays asymptotically as $\|\boldsymbol{k}\|^{-2\nu-p}$ Widom (1963), one has ($\nu = 3/2$)

$$\frac{1}{(2\pi)^p}\int_{-\pi}^\pi d^p\boldsymbol{\theta}\, e^{\pm i\boldsymbol{k}\cdot\boldsymbol{\theta}} \frac{1}{3\pi p}\sum_{i=1}^p |\theta_i|^3 \sim \sum_{i=1}^p k_i^{-4}\prod_{j\neq i}\delta_{k_j,0}, \quad \text{for } \|\boldsymbol{k}\| \to \infty \quad \text{(S42)}$$

and

$$\frac{1}{(2\pi)^p}\int_{-\pi}^\pi d^p\boldsymbol{\theta}\, e^{\pm i\boldsymbol{k}\cdot\boldsymbol{\theta}}\|\boldsymbol{\theta}\|^3 \sim \|\boldsymbol{k}\|^{-p-3}, \quad \text{for } \|\boldsymbol{k}\| \to \infty. \quad \text{(S43)}$$

All the other terms in the kernel expansion will result in subleading contributions in the Fourier transform. Therefore, the former of the two equations above yields the asymptotic scaling of eigenvalues of the local sector, whereas the latter yields the asymptotic scaling of the global sector.

The proof for the NTK case is analogous to the RFK case, except that the singular expansion near $\theta_i = 0$ is given by

$$\mathcal{K}_{\mathrm{NTK}}^{(2)}(\{\cos\theta_i\}_{i=1}^p) = 3 - \frac{1}{p}\sum_{i=1}^p \frac{|\theta_i|}{2} - \frac{\sqrt{2}}{\pi}\left(\frac{1}{p}\sum_{i=1}^p \frac{\theta_i^2}{2}\right)^{1/2} + \sum_{i=1}^p O(\theta_i^{3/2}). \quad \text{(S44)}$$

## C.3    Patches on the $s$-dimensional hypersphere

In this section, we make an additional step towards Thm. 3.1 by extending Thm. C.2 to the case of $s$-dimensional input patches. We still consider hierarchical kernels of depth 3 with the filter size of the second hidden layer set to $p = d/s$ (the total number of $s$-patches of the input) so as to ease the presentation. The extension to general depth and filter sizes is presented in Subsec. C.4.

**Theorem C.3 (Spectrum of depth-3 kernels on $s$-patches)** *Let $T_\mathcal{K}$ be the integral operator associated with a $d$-dimensional hierarchical kernel of depth 3, (2 hidden layers), with filter sizes $(s_1 = s, s_2)$ and $p_2 = 1$, such that $2s_2 = d$ and $s_2 = p$ (the number of $s$-patches). Eigenvalues and eigenfunctions of $T_\mathcal{K}$ can be organised into 2 sectors associated with the hidden layers of the kernel/network.*

  i. *The first sector consists of $s_1$-local eigenfunctions, which are functions of a single patch $\boldsymbol{x}_i$ for $i = 1, \ldots, p$. The labels $\boldsymbol{k}, \boldsymbol{\ell}$ of local eigenfunctions are such that all the $k_j$'s with $j \neq i$ are zero (because the eigenfunction is constant outside of $\boldsymbol{x}_i$). The corresponding eigenvalue is degenerate with respect to the location of the patch: we call it $\Lambda_{k_i}^{(1)}$. When $k_i \to \infty$,*
  $$\Lambda_{k_i}^{(1)} = \mathcal{C}_{s,1}\, k^{-2\nu-(s-1)} + o\left(k^{-2\nu-(s-1)}\right), \quad \text{(S45)}$$
  *with $\nu_{\mathrm{NTK}} = 1/2$, $\nu_{\mathrm{RFK}} = 3/2$. $\mathcal{C}_{s,1}$ can take two distinct strictly positive values depending on the parity of $k_i$;*

  ii. *The second sector consists of global eigenfunctions, which are functions of the whole input $\boldsymbol{x}$. The labels $\boldsymbol{k}, \boldsymbol{\ell}$ of global eigenfunctions are such that at least two of the $k_i$'s are nonzero. We call the corresponding eigenvalue $\Lambda_{\boldsymbol{k}}^{(2)}$. When $k \equiv \|\boldsymbol{k}\| \to \infty$, for fixed non-zero angles $\boldsymbol{k}/k$,*
  $$\Lambda_{\boldsymbol{k}}^{(2)} = \mathcal{C}_{s,2}\left(\frac{\boldsymbol{k}}{k}\right) k^{-2\nu-p(s-1)} + o\left(k^{-2\nu-p(s-1)}\right), \quad \text{(S46)}$$
  *where $\mathcal{C}_{s,2}$ is a positive function.*

*Proof.* A hierarchical RFK/NTK is a multi-dot-product kernel, therefore its eigenfunctions are products of spherical harmonics $\tilde{Y}_{\boldsymbol{k},\boldsymbol{\ell}}(\boldsymbol{x}) = \prod_{i=1}^{p} Y_{k_i,\ell_i}(\boldsymbol{x}_i)$ and the eigenvalues of $\mathcal{K}$ are given by Eq. S17,

$$\Lambda_{\boldsymbol{k}} = \left( \prod_{i=1}^{p} \frac{|\mathbb{S}^{s-2}|}{|\mathbb{S}^{s-1}|} \int_{-1}^{+1} dt_i \left(1 - t_i^2\right)^{\frac{s-3}{2}} P_{k_i,s}(t_i) \right) \mathcal{K}\left(\{t_i\}_i\right). \tag{S47}$$

The proof follows the following strategy: first, we show that the infinitely differentiable part of $\mathcal{K}$ results in eigenvalues which decay faster than any polynomial of the degrees $k_i$. We then show that the decay is controlled by the most singular term of the singular expansion of the kernel and finally compute such decay by relating it to the number of derivatives of the kernel having a finite l2 norm.

When $\mathcal{K}$ is infinitely differentiable in $[-1, +1]^p$, we can plug Rodrigues' formula Eq. S5 for each $P_{k_i,s}(t_i)$ and get

$$\Lambda_{\boldsymbol{k}} = \left( \prod_{i=1}^{p} \frac{|\mathbb{S}^{s-2}|}{|\mathbb{S}^{s-1}|} \left(-\frac{1}{2}\right)^{k_i} \frac{\Gamma\left(\frac{s-1}{2}\right)}{\Gamma\left(k_i + \frac{s-1}{2}\right)} \right) \int_{-1}^{+1} d\boldsymbol{t}\, \mathcal{K}\left(\boldsymbol{t}\right) \left( \prod_{i=1}^{p} \frac{d^{k_i}}{dt_i^{k_i}} \left(1 - t_i^2\right)^{k_i + \frac{s-3}{2}} \right), \tag{S48}$$

with $\int_{-1}^{+1} d\boldsymbol{t}$ denoting integration over the $p$-dimensional hypercube $[-1, +1]^p$. We can simplify the integral further via integration by parts, so as to obtain

$$\Lambda_{\boldsymbol{k}} = \left( \prod_{i=1}^{p} \frac{|\mathbb{S}^{s-2}|}{|\mathbb{S}^{s-1}|} \left(\frac{1}{2}\right)^{k_i} \frac{\Gamma\left(\frac{s-1}{2}\right)}{\Gamma\left(k_i + \frac{s-1}{2}\right)} \right) \int_{-1}^{+1} d\boldsymbol{t}\, \mathcal{K}^{(\boldsymbol{k})}\left(\boldsymbol{t}\right) \left( \prod_{i=1}^{p} \left(1 - t_i^2\right)^{k_i + \frac{s-3}{2}} \right), \tag{S49}$$

where $\mathcal{K}^{(\boldsymbol{k})}$ denotes the partial derivative of order $k_1$ with respect to $t_1$, $k_2$ with respect to $t_2$ and so on until $k_p$ with respect to $t_p$. Notice that the function $(1 - t^2)^{\frac{d-3}{2}}$ is proportional to the probability measure of the scalar product $t$ between two points sampled uniformly at random on the unit sphere (Atkinson & Han, 2012, Sec. 1.3),

$$|\mathbb{S}^{d-1}| = \int_{-1}^{+1} dt \left(1 - t^2\right)^{\frac{d-3}{2}} \int_{\mathbb{S}^{d-2}} dS^{d-2} \Rightarrow \frac{|\mathbb{S}^{d-1}|}{|\mathbb{S}^{d-2}|} \int_{-1}^{+1} dt \left(1 - t^2\right)^{\frac{d-3}{2}} = 1. \tag{S50}$$

This probability measure converges weakly to a Dirac mass $\delta(t)$ when $d \to \infty$. Recall, in addition, that $|\mathbb{S}^{d-1}| = 2\pi^{d/2}/\Gamma(d/2)$, where $\Gamma$ denotes the Gamma function $\Gamma(x) = \int_0^\infty dx\, x^{z-1} e^{-x}$. Thus, with converges weakly to a Dirac measure $\delta(t)$ as $c \to \infty$, once properly rescaled. In particular, choosing $k_i$ such that $k_i + (s-3)/2 = (d-3)/2$, one has

$$\lim_{k_i \to \infty} \frac{\Gamma\left(k_i + \frac{s}{2}\right)}{\sqrt{\pi}\Gamma\left(k_i + \frac{s-1}{2}\right)} \left(1 - t_i^2\right)^{k_i + \frac{s-3}{2}} = \delta(t_i). \tag{S51}$$

As a result, when $\mathcal{K}$ is infinitely differentiable, one has the following equivalence in the limit where all $k_i$'s are large,

$$\Lambda_{\boldsymbol{k}} \sim \left( \prod_{i=1}^{p} \frac{|\mathbb{S}^{s-2}|}{|\mathbb{S}^{s-1}|} \left(\frac{1}{2}\right)^{k_i} \frac{\Gamma\left(\frac{s-1}{2}\right)}{\Gamma\left(k_i + \frac{s}{2}\right)} \right) \mathcal{K}^{(\boldsymbol{k})}\left(\boldsymbol{0}\right), \tag{S52}$$

which implies that, when $\mathcal{K}$ is infinitely differentiable, the eigenvalues decay exponentially or faster with the $k_i$.

Let us now consider the nonanalytic part of $\mathcal{K}$. There are three kinds of terms appearing in the singular expansion of depth-3 kernels (cf. Subsec. C.1):

    *ia)* $c_+ \sum_i (1 - t_i)^\nu$ near $t_i = +1$;

    *ib)* $c_- \sum_i (1 + t_i)^\nu$ near $t_i = -1$;

    *ii)* $c_{+,\text{all}} \left(\sum_i (1 - t_i)/p\right)^\nu$ near $t_i = +1$ for all $i$;

where the exponent $\nu$ is $1/2$ for the NTK and $3/2$ for the RFK. We will not consider terms of the kind *ib)* explicitly, as the analysis is equivalent to that of terms of the kind *ia)*. After replacing $t_i$ with $\cos(\theta_i)$, as in Subsec. C.2, we get again $\sum_i |\theta_i|^{2\nu}$ and $\|\boldsymbol{\theta}\|^{2\nu}$ as leading nonanalytic terms. Therefore, we can rewrite the nonanalytic part of the kernel as follows,

$$\mathcal{K}_{\text{n.a.}}(\boldsymbol{\theta}) = \sum_i f_1(|\theta_i|) + f_2(\|\boldsymbol{\theta}\|) + \tilde{\mathcal{K}}(\boldsymbol{\theta}), \tag{S53}$$

where $f_1$, $f_2$ are single-variable functions which behave as $\theta^{2\nu}$ near zero and have compact support, whereas $\check{\mathcal{K}}$ has a singular expansion near $\theta_i = 0$ analogous to that of $\mathcal{K}$ but with leading nonanalyticities controlled by an exponent $\nu' \geq \nu + 1$.

Let us look at the contribution to the eigenvalue $\Lambda_{\boldsymbol{k}}$ due to the term $f_1(|\theta_i|)$:

$$
\left( \prod_{j=1}^{p} \frac{|\mathbb{S}^{s-2}|}{|\mathbb{S}^{s-1}|} \int_0^\pi d\theta_j \, (\sin(\theta_j))^{s-2} \, P_{k_j,s}(\cos(\theta_j)) \right) f_1(|\theta_i|)
$$

$$
= \left( \prod_{\substack{j \neq i}} \delta_{k_j,0} \right) \frac{|\mathbb{S}^{s-2}|}{|\mathbb{S}^{s-1}|} \int_0^\pi d\theta \, (\sin(\theta))^{s-2} \, P_{k_i,s}(\cos(\theta)) f_1(|\theta|) = \left( \prod_{\substack{j \neq i}} \delta_{k_j,0} \right) (f_1)_{k_1} ,
$$

(S54)

where we have introduced $(f_1)_k$ as the projection of $f_1(\theta)$ on the $k$-th Legendre polynomial. The asymptotic decay of $(f_1)_k$ is strictly related to the differentiability of $f_1$, which is in turn controlled by action of the Laplace-Beltrami operator $\Delta$ on $f_1$. As a function on the sphere $\mathbb{S}^{s-1}$, $f_1$ depends only on one angle, therefore the Laplace-Beltrami operator acts as follows,

$$
\Delta f_1(\theta) = \frac{1}{\sin(\theta)^{s-2}} \frac{d}{d\theta} \left( \sin(\theta)^{s-2} \frac{df_1}{d\theta}(\theta) \right) = f_1''(\theta) + (d-2) \frac{\cos(\theta)}{\sin(\theta)} f_1'(\theta).
$$

(S55)

In terms of singular behaviour near $\theta = 0$, $f_1(\theta) \sim |\theta|^{2\nu}$ implies $\Delta f_1(\theta) \sim |\theta|^{2\nu-2}$, thus $\Delta^m f_1(\theta) \sim |\theta|^{2(\nu-m)}$. Given $\nu$, repeated applications of $\Delta$ eventually result in a function whose l2 norm on the sphere diverges. On the one hand,

$$
\|\Delta^{m/2} f_1\|^2 = \int_0^\pi d\theta \, \sin^{d-2}(\theta) f_1(\theta) \Delta^m f_1(\theta).
$$

(S56)

The integrand behaves as $|\theta|^{d-2+4\nu-2m}$ near 0, thus the integral diverges for $m \geq 2\nu + (d-1)/2$. On the other hand, from Eq. S3,

$$
\|\Delta^{m/2} f_1\|^2 = \sum_k \mathcal{N}_{k,s} \left( k(k+s-2) \right)^m |(f_1)_k|^2.
$$

(S57)

As $\mathcal{N}_{k,s} \sim k^{s-2}$ and the sum must converge for $m < 2\nu + (d-1)/2$ and diverge otherwise, $(f_1)_k \sim k^{-2\nu-(s-1)}$. The projections of all the other terms in $\mathcal{K}$ on Legendre polynomials of one of the $p$ angles $\theta_i$ display a faster decay with $k$, therefore the above results imply the asymptotic scaling of local eigenvalues. Notice that such scaling matches with the result of Bietti & Bach (2021), which was obtained with a different argument.

Finally, let us look at the contribution to the eigenvalue $\Lambda_{\boldsymbol{k}}$ due to the term $f_2(\|\boldsymbol{\theta}\|)$:

$$
\left( \prod_{j=1}^{p} \frac{|\mathbb{S}^{s-2}|}{|\mathbb{S}^{s-1}|} \int_0^\pi d\theta_j \, (\sin(\theta_j))^{s-2} \, P_{k_j,s}(\cos(\theta_j)) \right) f_2(\|\boldsymbol{\theta}\|) = (f_2)_{\boldsymbol{k}} ,
$$

(S58)

where we have introduced $(f_2)_{\boldsymbol{k}}$ as the projection of $f_2(\|\boldsymbol{\theta}\|)$ on the multi-Legendre polynomial with multi-degree $\boldsymbol{k}$. The asymptotic decay of $(f_2)_{\boldsymbol{k}}$ is again related to the differentiability of $f_2$, controlled by action of the multi-sphere Laplace-Beltrami operator $\Delta_{p,s}$ in Eq. S13. As $f_2$ depends only on one angle per sphere,

$$
\Delta_{p,s} f_2(\|\boldsymbol{\theta}\|) = \sum_{i=1}^{p} \left( \partial_{\theta_i}^2 f_2(\|\boldsymbol{\theta}\|) + (s-2) \frac{\cos(\theta_i)}{\sin(\theta_i)} \partial_{\theta_i} f_2(\|\boldsymbol{\theta}\|) \right).
$$

(S59)

Further simplifications occur since $f_2$ depends only on the norm of $\boldsymbol{\theta}$. In terms of the singular behaviour near $\|\boldsymbol{\theta}\| = 0$, $f_2 \sim \|\boldsymbol{\theta}\|^{2\nu}$ implies $\Delta_{p,s}^m f_2 \sim \|\boldsymbol{\theta}\|^{2(\nu-m)}$, thus

$$
\|\Delta_{p,s}^{m/2} f_2\|^2 = \int_{[0,\pi]^p} d^p\boldsymbol{\theta} \prod_{i=1}^{p} \left( \sin^{s-2}(\theta_i) \right) f_2(\|\boldsymbol{\theta}\|) \Delta_{p,s}^m f_2(\|\boldsymbol{\theta}\|) < +\infty
$$

(S60)

requires $m < 2\nu + p(s-1)/2$ (compare with $m < 2\nu + (s-1)/2$ for the local contributions). Therefore, one has

$$\|\Delta_{p,s}^{m/2} f_1\|^2 = \sum_{\boldsymbol{k}} \left(\prod_{i=1}^{p} \mathcal{N}_{k_i,s}\right) \left(\sum_{i=1}^{p} k_i(k_i + s - 2)\right)^m |(f_2)_{\boldsymbol{k}}|^2 < +\infty \quad \forall\, m < 2\nu + p(s-1)/2, \tag{S61}$$

while the sum diverges for $m \geq 2\nu + p(s-1)/2$. In addition, since $f_2$ is a radial function of $\boldsymbol{\theta}$ which is homogeneous (or scale-invariant) near $\|\boldsymbol{\theta}\| = 0$, $(f_2)_{\boldsymbol{k}}$ can be factorised in the large-$\|\boldsymbol{k}\|$ limit into a power of the norm $\|\boldsymbol{k}\|^{\alpha}$ and a finite angular part $\mathcal{C}(\boldsymbol{k}/\|\boldsymbol{k}\|)$. By plugging the factorisation into Eq. S61, we get

$$(f_2)_{\boldsymbol{k}} \sim \mathcal{C}(\boldsymbol{k}/\|\boldsymbol{k}\|)\|\boldsymbol{k}\|^{-2\nu-p(s-1)}, \quad \sum_{\boldsymbol{k},\|\boldsymbol{k}\|=k} \left(\left(\prod_{i=1}^{p}(k_i/k)^{s-2}\right) \mathcal{C}(\boldsymbol{k}/\|\boldsymbol{k}\|)^2\right) < +\infty \tag{S62}$$

The projections of all the other terms in $\mathcal{K}$ on multi-Legendre polynomials display a faster decay with $\|\boldsymbol{k}\|$, therefore the above results imply the asymptotic scaling of global eigenvalues.

## C.4  GENERAL DEPTH

The generalisation to arbitrary depth is trivial once the depth-3 case is understood. For global and $s_1$-local eigenvalues, the analysis of the previous section carries over unaltered. All the other intermediate sectors correspond to the other terms singular expansion of the kernel: from Subsec. C.1, these terms can be written as

$$\frac{c}{p_L} \frac{1}{\left(\prod_{l' < l'' \leq L} s_{l''}\right)} \sum_{i_{l'+1 \to L+1}} \left(\frac{1}{\left(\prod_{2 \leq l'' \leq l'} s_{l''}\right)} \sum_{i_{2 \to l'}} \left(1 - t_{i_{2 \to L+1}}\right)\right)^{\nu}, \tag{S63}$$

for some $l' = 2, \ldots, L - 1$ and fractional $\nu$. In practice, this term is a sum over the $p_{l'} = p_L \prod_{l' < l'' \leq L} s_{l''}$ meta-patches of $\boldsymbol{t}$ having size $s_{2 \to l'} := \prod_{2 \leq l'' \leq l'} s_{l''}$. Each summand is the fractional power $\nu$ of the average of the $t_i$'s within a meta-patch. When plugging such term into Eq. S47, the integrals over the $t_i$'s which do not belong to that meta-patch yield Kronecker deltas for the corresponding $k_i$'s. The integrals over the $t_i$'s within the meta-patch, instead, can be written as in Eq. S58 with the product and the norm restricted over the elements of that meta-patch, i.e., $\|\boldsymbol{\theta}\| \to \left(\sum_{i_{2 \to l'}} \theta_{i_{2 \to L+1}}^2\right)^{1/2}$. Therefore, the scaling of the eigenvalue with $k$ is given again by Eq. S63, but with $p$ replaced by the size of the meta-patch $\prod_{2 \leq l'' \leq l'} s_{l''}$, so that the effective dimension of Eq. 3 appears at the exponent.

## D  GENERALISATION BOUNDS FOR KERNEL REGRESSION AND SPATIAL ADAPTIVITY

This appendix provides an introduction to classical generalisation bounds for kernel regression and extends Cor. 4.1 to patches on the hypersphere.

## D.1  CLASSICAL GENERALISATION BOUNDS

Consider the regression setting detailed in Sec. 4 of the main text. First, assume that the target function $f^*$ belongs to the RKHS $\mathcal{H}$ of the kernel $\mathcal{K}$. Then, without further assumptions on $\mathcal{K}$, we have the following dimension-free bound on the excess risk, based on Rademacher complexity (Bach, 2021, Chs. 4, 7), Bietti (2022),

$$\bar{\epsilon}(\lambda, n) - \epsilon(f^*) \leq \mathcal{C} \, \|f^*\|_{\mathcal{H}} \sqrt{\frac{\mathrm{Tr}(\mathcal{T}_{\mathcal{K}})}{n}}, \tag{S64}$$

where $\mathcal{T}_\mathcal{K}$ is the integral operator associated to $\mathcal{K}$. For a hierarchical kernel, having a target with more power in the local sectors can result in a smaller $\|f^*\|_\mathcal{H}$, hence a smaller excess risk. However, this gain is only a constant factor in terms of sample complexity and, more importantly, being in the RKHS requires an order of smoothness which typically is of the order of the dimension, which is a very-restrictive assumption in high-dimensional settings. This result can be extended by including more details about the kernel and the target function.

In particular, (Bach, 2021, Prop. 7.2) states that, for $f^*$ in the closure of $\mathcal{H}$, regularisation $\lambda \le 1$ and $n \ge \frac{5}{\lambda}(1 + \log(1/\lambda))$, one has

$$\bar{\epsilon}(\lambda, n) - \epsilon(f^*) \le 16\,\frac{\sigma^2}{n}\,\mathrm{Tr}\left((\mathcal{T}_\mathcal{K} + \lambda I)^{-1}\mathcal{T}_\mathcal{K}\right) + 16\inf_{f \in \mathcal{H}}\left\{\|f - f^*\|_{L_2}^2 + \lambda\|f\|_\mathcal{H}^2\right\} + \frac{24}{n^2}\,\|f^*\|_{L_\infty},$$
(S65)

where $\sigma^2$ bounds the conditional variance of the labels, i.e. $\mathbb{E}_{(\boldsymbol{x},y)\sim p}\left[(y - f^*(\boldsymbol{x}))^2 \mid \boldsymbol{x}\right] < \sigma^2$.

Then, let us consider the following standard assumptions in the kernel literature Caponnetto & De Vito (2007),

$$\text{capacity: } \mathrm{Tr}\left(\mathcal{T}_\mathcal{K}^{1/\alpha}\right) = \sum_{\boldsymbol{k}\ge 0}\sum_{\boldsymbol{\ell}}(\Lambda_{\boldsymbol{k}})^{1/\alpha} < +\infty,$$

$$\text{source: } \left\|T_\mathcal{K}^{\frac{1-r}{2}}f^*\right\|_\mathcal{H}^2 = \sum_{\boldsymbol{k}\ge 0}\sum_{\boldsymbol{\ell}}(\Lambda_{\boldsymbol{k}})^{-r}(f_{\boldsymbol{k},\boldsymbol{\ell}}^*)^2 < +\infty.$$
(S66)

In short, the first assumption characterises the 'size' of the RKHS (the larger $\alpha$, the smaller the number of functions in the RKHS), while the second assumption defines the regularity of the target function relative to that of the kernel (when $r = 1$, $f^* \in \mathcal{H}$; when $r < 1$, $f^*$ is less smooth; when $r > 1$, $f^*$ is smoother). Combining these assumptions with Eq. S65, one gets

$$\bar{\epsilon}(\lambda, n) - \epsilon(f^*) \le 16\,\frac{\sigma^2}{n}\,\mathcal{C}_1\lambda^{-1/\alpha} + 16\,\mathcal{C}_2\,\lambda^r + \frac{24}{n^2}\,\|f^*\|_{L_\infty}.$$
(S67)

Optimising for $\lambda$ results in

$$\lambda_n = \left(\frac{\mathcal{C}_1\sigma^2}{\alpha\,r\,\mathcal{C}_2\,n}\right)^{\frac{\alpha}{\alpha r + 1}},$$
(S68)

and the bound becomes

$$\bar{\epsilon}(\lambda_n, n) - \epsilon(f^*) \lesssim \mathcal{C}_2^{\frac{2}{\alpha r + 1}}\left(\frac{\mathcal{C}_1\sigma^2}{n}\right)^{\frac{\alpha r}{\alpha r + 1}} + \frac{1}{n^2}\,\|f^*\|_{L_\infty}.$$
(S69)

Finally, when $r > (\alpha - 1)/\alpha$, $n \ge \frac{5}{\lambda_n}(1 + \log(1/\lambda_n))$ is always satisfied for $n$ large enough.

### D.2 Extension of Cor. 4.1 to patches on the hypersphere

**Corollary D.1 (Adaptivity to spatial structure)** *Let $T_\mathcal{K}$ be the integral operator of the kernel of a hierarchical deep CNN as in Thm. 3.1. Then:* i) *the* capacity *exponent $\alpha$ is controlled by the largest sector of the spectrum, i.e.*

$$\mathrm{Tr}\left(\mathcal{T}_\mathcal{K}^{1/\alpha}\right) < +\infty \Leftrightarrow \alpha < 1 + 2\nu/d_{\mathrm{eff}}(L);$$
(S70)

ii) *the* source *exponent $r$ is controlled by the structure of the target function $f^*$, i.e., if there is $l \le L$ such that $f^*$ depends only on some meta-patch $\boldsymbol{x}_{i_{l+1}\to L+1}$, then only the first $l$ sectors of the spectrum contribute to the source condition,*

$$\left\|T_\mathcal{K}^{\frac{1-r}{2}}f^*\right\|_\mathcal{H}^2 = \sum_{l'=1}^{l}\sum_{\substack{i_{l'+1}\to L+1}}\sum_{\substack{\boldsymbol{k}_{i_{l'+1}\to L+1}\\ \boldsymbol{\ell}_{i_{l'+1}\to L+1}}}\left(\Lambda_{\boldsymbol{k}_{i_{l'+1}\to L+1}}^{(l')}\right)^{-r}\left(f_{\boldsymbol{k}_{i_{l'+1}\to L+1},\boldsymbol{\ell}_{i_{l'+1}\to L+1}}^*\right)^2.$$
(S71)

*The same holds if $f^*$ is a linear combination of such functions. As a result, when $d_{\mathrm{eff}}(L)$ is large and $\alpha \to 1$, the decay of the error is controlled by the effective dimensionality of the target $d_{\mathrm{eff}}(l)$.*

### D.3 DECAY OF EIGENVALUES WITH THE RANK

**Shallow kernels.** Consider a depth-two kernel with filters of size $s$. Our goal is to compute the scaling of the eigenvalues of the kernel $\Lambda_{\boldsymbol{k}}$ with their rank $\rho$. The eigenvalues decay with $\boldsymbol{k}$ as

$$\Lambda_{\boldsymbol{k}} \sim \sum_{i=1}^{p} k_i^{-2\nu_S-(s-1)} \prod_{j\neq i} \delta_{k_j,0}. \tag{S72}$$

In order to take into account their algebraic multiplicity, we introduce the eigenvalue density $\mathcal{D}(\Lambda)$, whose asymptotic form for small eigenvalues is

$$
\begin{aligned}
\mathcal{D}(\Lambda) &= \sum_{\boldsymbol{k},\boldsymbol{\ell}} \delta(\Lambda - \Lambda_{\boldsymbol{k}}) \\
&\sim \sum_{\boldsymbol{k}} \left( \prod_{i=1}^{p} k_i^{s-2} \right) \delta \left( \Lambda - \sum_{i=1}^{p} k_i^{-2\nu-(s-1)} \prod_{j\neq i} \delta_{k_j,0} \right) \\
&\sim \sum_{i=1}^{p} \sum_{k_i} k_i^{s-2} \delta \left( \Lambda - k_i^{-2\nu-(s-1)} \right) \\
&\sim \int_1^{\infty} dk \, k^{s-2} \delta \left( \Lambda - k^{-2\nu-(s-1)} \right) \\
&\sim \Lambda^{-1-\frac{s-1}{2\nu+(s-1)}}.
\end{aligned}
\tag{S73}
$$

Thus, the scaling of $\Lambda(\rho)$ can be determined self-consistently,

$$\rho = \int_{\Lambda(\rho)}^{\Lambda(1)} d\Lambda \, \mathcal{D}(\Lambda) \sim \Lambda(\rho)^{-\frac{s-1}{s\nu+(s-1)}} \implies \Lambda(\rho) \sim \rho^{-1-\frac{2\nu}{s-1}}. \tag{S74}$$

**Deep kernels.** Consider a kernel of depth $L+1$ with filter sizes $(s_1, \ldots, s_L)$ and $p_L = 1$. For each sector $l$, one can compute the density of eigenvalues $\mathcal{D}_{(l)}(\Lambda)$. Depending on $s_1$, there are two different cases.

If $s_1 = 2$,

$$
\begin{aligned}
\mathcal{D}_{(l)}(\Lambda) &= \sum_{\boldsymbol{k}} \delta(\Lambda - \Lambda_{\boldsymbol{k}}^{(l)}) \\
&\sim \sum_{i_{l+1\to L+1}} \sum_{\boldsymbol{k}_{i_{l+1\to L+1}}} \delta \left( \Lambda - \mathcal{C}_{2,l} \, \|\boldsymbol{k}_{i_{l+1\to L+1}}\|^{-2\nu-d_{\mathrm{eff}}(l)} \right) \\
&\sim \int_1^{\infty} dk \, k^{d_{\mathrm{eff}}(l)-1} \delta \left( \Lambda - \mathcal{C}_{2,l} \, k^{-2\nu-d_{\mathrm{eff}}(l)} \right) \\
&\sim \Lambda^{-1-\frac{d_{\mathrm{eff}}(l)}{2\nu+d_{\mathrm{eff}}(l)}}.
\end{aligned}
\tag{S75}
$$

If $s_1 \geq 3$,

$$
\begin{aligned}
\mathcal{D}_{(l)}(\Lambda) &= \sum_{\boldsymbol{k},\boldsymbol{\ell}} \delta(\Lambda - \Lambda_{\boldsymbol{k}}^{(l)}) \\
&\sim \sum_{\substack{i_{l+1\to L+1} \boldsymbol{k}_{i_{l+1\to L+1}}, \\ \boldsymbol{\ell}_{i_{l+1\to L+1}}}} \delta \left( \Lambda - \mathcal{C}_{s_1,l} \left( \frac{\boldsymbol{k}_{i_{l+1\to L+1}}}{\|\boldsymbol{k}_{i_{l+1\to L+1}}\|} \right) \|\boldsymbol{k}_{i_{l+1\to L+1}}\|^{-2\nu-d_{\mathrm{eff}}(l)} \right) \\
&\sim \Lambda^{-1-\frac{d_{\mathrm{eff}}(l)}{2\nu+d_{\mathrm{eff}}(l)}}.
\end{aligned}
\tag{S76}
$$

When summing over all layers $l$'s, the asymptotic behaviour of the total density of eigenvalues $\mathcal{D}(\Lambda) = \sum_l \mathcal{D}_{(l)}(\Lambda)$ is dictated by the density of the sector with the slowest decay, i.e. the last one. Hence,

$$\mathcal{D}(\Lambda) \sim \Lambda^{-1-\frac{d_{\mathrm{eff}}(L)}{2\nu+d_{\mathrm{eff}}(L)}}. \tag{S77}$$

Therefore, similarly to the shallow case, one finds self-consistently that the $\rho$-th eigenvalue of the kernel decays as

$$\Lambda(\rho) \sim \rho^{-1-\frac{2\nu}{d_{\text{eff}}(L)}}. \tag{S78}$$

# E    STATISTICAL MECHANICS OF GENERALISATION IN KERNEL REGRESSION

In Bordelon et al. (2020); Canatar et al. (2021), the authors derived a heuristic expression for the average-case mean-squared error of kernel (ridge) regression with the replica method of statistical physics Mézard et al. (1987). Denoting with $\{\phi_\rho(\boldsymbol{x}), \Lambda_\rho\}_{\rho \geq 1}$ the eigenfunctions and eigenvalues of the kernel and with $c_\rho$ the coefficients of the target function in this basis, i.e. $f^*(\boldsymbol{x}) = \sum_{\rho \geq 1} c_\rho \phi_\rho(\boldsymbol{x})$, one has

$$\epsilon(\lambda, n) = \partial_\lambda \left( \frac{\kappa_\lambda(n)}{n} \right) \sum_\rho \frac{\kappa_\lambda(n)^2}{(n\Lambda_\rho + \kappa_\lambda(n))^2} \, \mathbb{E}[c_\rho^2], \tag{S79}$$

where $\lambda$ is the ridge and $\kappa(n)$ satisfies the implicit equation

$$\frac{\kappa_\lambda(n)}{n} = \lambda + \frac{1}{n} \sum_\rho \frac{\Lambda_\rho \kappa_\lambda(n)/n}{\Lambda_\rho + \kappa_\lambda(n)/n}. \tag{S80}$$

In short, the replica calculation used to obtain these equations consists in defining an energy functional $\mathcal{E}(f)$ related to the empirical MSE and assigning to the predictor $f$ a Boltzmann measure, i.e. $P(f) \propto e^{-\beta E(f)}$. When $\beta \to \infty$, the measure concentrates around the minimum of $\mathcal{E}(f)$, which coincides with the minimiser of the empirical MSE. Then, since $\mathcal{E}(f)$ depends only quadratically on the projections $c_\rho$, computing the average over data that appears in the definition of the generalisation error, reduces to computing Gaussian integrals. While non-rigorous, this method has been successfully used in physics—to study disordered systems—and in machine learning theory. In particular, the predictions obtained with Eq. S79 and Eq. S80 have been validated numerically for both synthetic and real datasets.

In Eq. S79, $\kappa_\lambda(n)/n$ plays the role of a threshold: the modal contributions to the error tend to $0$ for $\rho$ such that $\Lambda_\rho \gg \kappa_\lambda(n)/n$, and to $\mathbb{E}[c_\rho^2]$ for $\rho$ such that $\Lambda_\rho \ll \kappa_\lambda(n)/n$. This is equivalent to saying that kernel regression can capture only the modes corresponding to the eigenvalues larger than $\kappa_\lambda(n)/n$ (see also Jacot et al. (2020a;b)).

In the ridgeless limit $\lambda \to 0^+$, this threshold asymptotically tends to the $n$-th eigenvalue of the student, resulting in the intuitive picture presented in the main text. Namely, given $n$ training points, ridgeless regression learns the $n$ projections corresponding to the highest eigenvalues. In particular, assume that the kernel spectrum and the target function projections decay as power laws. Namely, $\Lambda_\rho \sim \rho^{-a}$ and $\mathbb{E}[c_\rho^2] \sim \rho^{-b}$, with $2a > b - 1$. Furthermore, we can approximate the summations over modes with an integral by using the Euler-MacLaurin formula. Hence, we substitute the eigenvalues with their asymptotic limit $\Lambda_\rho = A\rho^{-a}$. Since, $\kappa_0(n)/n \to 0$ as $n \to \infty$, these two operations result in an error which is asymptotically independent of $n$. In particular,

$$\frac{\kappa_0(n)}{n} = \frac{\kappa_0(n)}{n} \frac{1}{n} \left( \int_0^\infty \frac{A\rho^{-a}}{A\rho^{-a} + \kappa_0(n)/n} \, d\rho + O(1) \right)$$

$$= \frac{\kappa_0(n)}{n} \frac{1}{n} \left( \left( \frac{\kappa_0(n)}{n} \right)^{-\frac{1}{a}} \int_0^\infty \frac{\sigma^{\frac{1}{a}-1} A^{\frac{1}{a}} a^{-1}}{1+\sigma} \, d\sigma + O(1) \right). \tag{S81}$$

Since the integration over $\sigma$ is finite and independent of $n$, we obtain that $\kappa_0(n)/n = O(n^{-a})$. Similarly, we find that the mode-independent prefactor $\partial_\lambda \left( \kappa_\lambda(n)/n \right)|_{\lambda=0} = O(1)$.

As a result, we have

$$\epsilon(n) \sim \sum_\rho \frac{n^{-2a}}{(A\rho^{-a} + n^{-a})^2} \, \mathbb{E}[c_\rho^2]. \tag{S82}$$

Following the intuitive argument about the thresholding action of $\kappa_0(n)/n \sim n^{-a}$, we can split the summation in Eq. S82 into modes where $\Lambda_\rho \gg \kappa_0(P)/n$, $\Lambda_\rho \sim \kappa_0(n)/n$ and $\Lambda_\rho \ll \kappa_0(n)/n$,

$$\epsilon(n) \sim \sum_{\rho \ll n} \frac{n^{-2a}}{(A\rho^{-a})^2} \mathbb{E}[c_\rho^2] + \sum_{\rho \sim n} \frac{1}{2} \mathbb{E}[c_\rho^2] + \sum_{\rho \gg n} \mathbb{E}[c_\rho^2]. \tag{S83}$$

Finally, Eq. 12 is obtained by noticing that, under the assumption on the decay of $\mathbb{E}[c_\rho^2]$, the contribution of the summation over $\rho \ll n$ is subleading in $n$, whereas the other two can be merged together.

## F EXAMPLES

### F.1 RATES FROM SPECTRAL BIAS ANSATZ

Consider a target function $f^*$ which only depends on the meta-patch $\boldsymbol{x}_{i_{l+1\to L+1}}$ and with square-integrable derivatives up to order $m$, i.e. $\|\Delta^{m/2} f^*\|^2 < +\infty$, with $\Delta$ denoting the Laplace operator. Moreover, consider a hierarchical kernel of depth $L+1$ with filter sizes $(s_1, \ldots, s_L)$ and $p_L = 1$. We want to compute the asymptotic scaling of the error by using Eq. 12, i.e.

$$\overline{\epsilon}(n) \sim \sum_{\boldsymbol{k},\boldsymbol{\ell} \text{ s.t. } \Lambda_{\boldsymbol{k}} < \Lambda(n)} |f^*_{\boldsymbol{k},\boldsymbol{\ell}}|^2. \tag{S84}$$

In the previous section, we showed that the $n$-th eigenvalue of the kernel $\Lambda(n)$ decays as

$$\Lambda(n) \sim n^{-1 - \frac{2\nu}{d_{\text{eff}}(L)}}. \tag{S85}$$

Since by construction the target function depends only on a meta-patch of the $l$-th sector, the only non-zero projections will be the ones on eigenfunctions of the first $l$ sectors. Thus, all the $\boldsymbol{k}$'s corresponding to the sectors of layers with $l' > l$ do not contribute to the sum. In particular, the sum is dominated by the $\boldsymbol{k}$'s of the largest sector and the set $\{\boldsymbol{k} \text{ s.t. } \Lambda_{\boldsymbol{k}} < \Lambda(n)\}$ is the set of $\boldsymbol{k}_{i_{l+1\to L+1}}$'s with norm larger than $n^{\frac{2\nu + d_{\text{eff}}(L)}{(2\nu + d_{\text{eff}}(l))\, d_{\text{eff}}(L)}}$.

Finally, we notice that the finite-norm condition on the derivatives,

$$\|\Delta^{m/2} f^*\|^2 = \sum_{\boldsymbol{k}} \left(\prod_{i=1}^{p} \mathcal{N}_{k_i,s}\right) \left(\sum_{i=1}^{p} k_i(k_i + s - 2)\right)^m |f^*_{\boldsymbol{k},\boldsymbol{\ell}}|^2 < +\infty, \tag{S86}$$

implies $|f^*_{\boldsymbol{k},\boldsymbol{\ell}}|^2 \lesssim \|\boldsymbol{k}\|^{-2m - d_{\text{eff}}(L)}$ (see Subsec. C.3).

Hence, plugging everything in Eq. S84 we find

$$\overline{\epsilon}(n) \sim n^{-\frac{2m}{2\nu + d_{\text{eff}}(l)} \frac{2\nu + d_{\text{eff}}(L)}{d_{\text{eff}}(L)}}. \tag{S87}$$

## G NUMERICAL EXPERIMENTS

### G.1 EXPERIMENTAL SETUP

Experiments were run on a high-performance computing cluster with nodes having Intel Xeon Gold processors with 20 cores and 192 GB of DDR4 RAM. All codes are written in PyTorch Paszke et al. (2019).

### G.2 TEACHER-STUDENT LEARNING CURVES

In order to obtain the learning curves, we generate $n + n_{\text{test}}$ random points uniformly distributed on the product of hyperspheres over the patches. We use $n \in \{128, 256, 512, 1024, 2048, 4096, 8192\}$ and $n_{\text{test}} = 8192$. For each value of $n$, we sample a Gaussian random field with zero mean and covariance given by the teacher kernel. Then, we compute the kernel regression predictor of the student kernel, and we estimate the generalisation error as the mean squared error of the obtained predictor on the $n_{\text{test}}$ unseen example. The expectation over the teacher randomness is obtained by averaging over 16 independent sets of random input points and realisations of the Gaussian random fields. As teacher and student kernels, we use the analytical forms of the neural tangent kernels of hierarchical convolutional networks, with different combinations of depths and filter sizes.

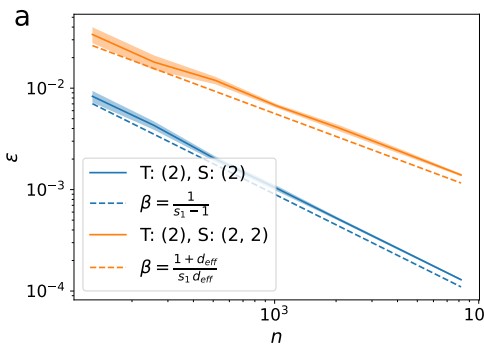 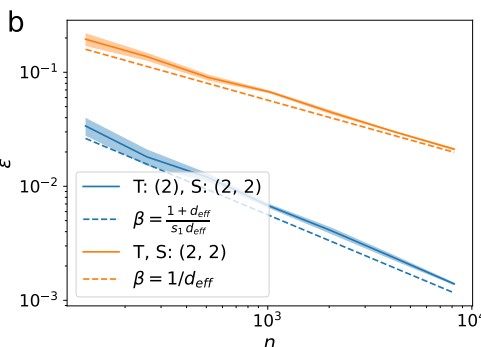

Figure S1: Learning curves for deep convolutional NTKs ($\nu = 1/2$) in a teacher-student setting. **a.** Depth-two teachers learned by depth-two (matched) and depth-three (mismatched) students. Both these students are not cursed by the input dimension. **b.** Depth-three students learning depth-two and depth-three teachers. These students are cursed only in the second case. The numbers inside brackets are the sequence of filter sizes of the kernels. Solid lines are the results of experiments averaged over 16 realisations with the shaded areas representing the empirical standard deviations. The predicted asymptotic scaling $\epsilon \sim n^{-\beta}$ are reported as dashed lines.

**Depth-two and depth-three architectures.** Fig. S1 reports the learning curves of depth-two and depth-three kernels with binary filters at all layers. Depth-three students defeat the curse of dimensionality when learning depth-two teachers, achieving a similar performance of depth-two students matched to the teacher's structure. However, as we predict, these students encounter the curse of dimensionality when learning depth-three teachers.

**Ternary filters.** Fig. S2 reports the learning curves for kernels with 3-dimensional filters and confirms our predictions in the $s_1 \geq 3$ case.

**Comparison with the noisy and optimally-regularised case.** Panel (a) of Fig. S3 compares the learning curves obtained in the optimally-regularised and ridgeless cases for noisy and noiseless data, respectively. The first case corresponds to the setting studied in Caponnetto & De Vito (2007), in which the source-capacity formalism applies. In contrast with the second setting—which is the one used in the teacher-student scenarios and where it holds the correspondence between kernel methods and neural networks—*i)* we add to the labels a Gaussian random noise with standard deviation $\sigma = 0.1$, *ii)* for each $n$, we select the ridge resulting in the best generalisation performance. We observe that the decay obtained in the bound derived from the source-capacity conditions is exactly the one found numerically, i.e. the rate of the bound is tight. As a further check, panel (b) shows that the optimal ridge decays as prescribed.

G.3    Illustration of different teacher-student scenarios

In this subsection, we comment on the results obtained in the different teacher-student scenarios of Fig. 2, panel (a), and Fig. S1, panel (a). To ease notation, in the following we always consider the NTK for both teacher and student kernels, i.e. smoothness exponent $\nu_T = \nu_S = 1/2$. However, we point out that when the teacher kernel is a hierarchical RFK ($\nu_T = 3/2$), the target function corresponds to the output of an infinitely-wide, deep hierarchical network at initialisation[4]. The error rates are obtained from Eq. 17, after setting the smoothness exponent $m = \nu_T$ (the smoothness exponent of the teacher covariance kernel).

---

[4]See, e.g, Lee et al. (2017) for the equivalence between infinitely-wide networks and Gaussian random fields with covariance given by the RFK.

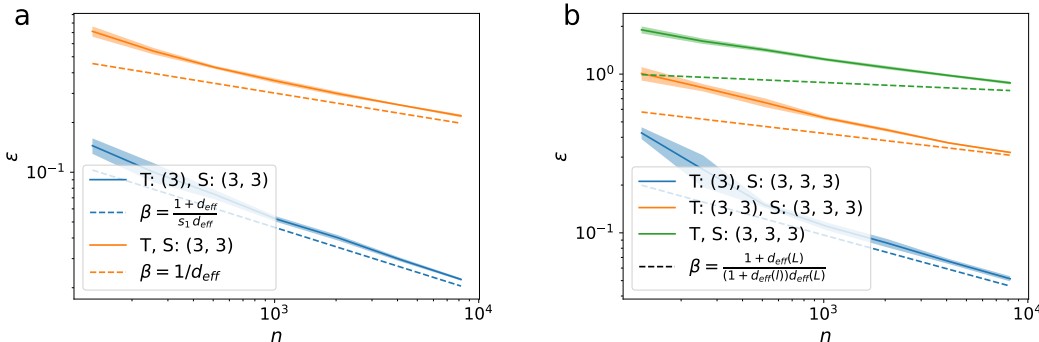

Figure S2: Learning curves for deep convolutional NTKs ($\nu = 1/2$) with filters of size 3 in a teacher-student setting. **a.** Depth-three students learning depth-two and depth-three teachers. These students are cursed only in the second case. **b.** Depth-three models are cursed by the effective input dimensionality. The numbers inside brackets are the sequence of filter sizes of the kernels. Solid lines are the results of experiments averaged over 16 realisations with the shaded areas representing the empirical standard deviations. The predicted asymptotic scaling $\epsilon \sim n^{-\beta}$ are reported as dashed lines.

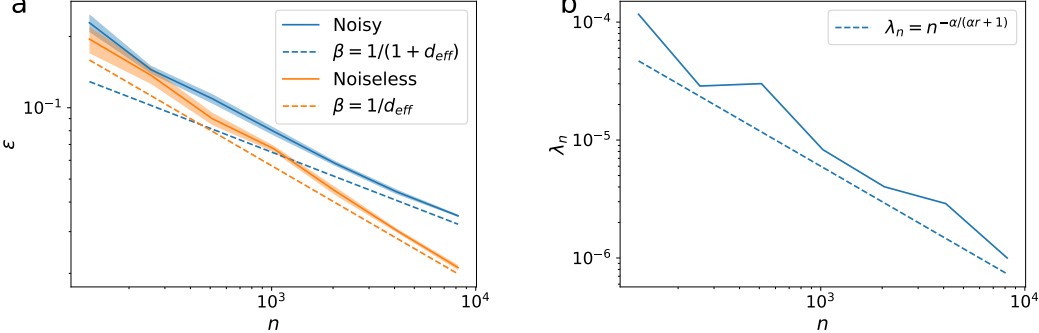

Figure S3: Noisy (optimally-regularised) vs noiseless (ridgeless) learning curves for depth-three deep convolutional NTKs ($\nu = 1/2$) in a teacher-student setting. **a.** Comparison between the learning curves in the noisy and noiseless case. Dashed lines represent the rates predicted with source-capacity bounds and replica calculations, respectively. Shaded areas represent the empirical standard deviations. **b.** Decay of the optimal ridge with the number of training points.

The first case we consider consists of one-hidden-layer convolutional teacher (left) and student (right) kernels.

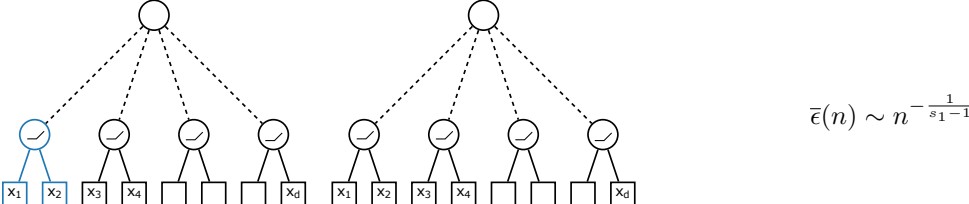

$$\bar{\epsilon}(n) \sim n^{-\frac{1}{s_1 - 1}}$$

As highlighted in blue, the output of the teacher is a linear combination (dashed lines indicate the linear output weights) of $s_1$-dimensional functions of the input patches. If the structure of the student is matched to the one of the teacher, the learning problem becomes effectively $(s_1 - 1)$-dimensional and the error decays as $n^{-1/(s_1-1)}$, instead of $n^{-1/d_{\text{eff}}}$, with $d_{\text{eff}}$ the total input dimension with the number of spherical constraints subtracted (one per patch). Notice that the role of the student's structure, i.e. the algorithm, is as crucial as the role of the teacher, i.e. the task. Indeed, using a fully-connected student with no prior on the task's locality would result in an error's decay cursed by dimensionality. However, in contrast to fully-connected students, shallow convolutional students are only able to learn tasks with the same structure. In particular, any task entailing non-linear interactions between patches—which are arguably crucial in order to learn image data—belongs to their null space.

As we illustrated in the main text, to solve this strong constraint on the hypothesis space, one has to consider deep convolutional architectures. In particular, consider the same shallow teacher of the previous paragraph (left) learnt by a depth-four convolutional student (right).

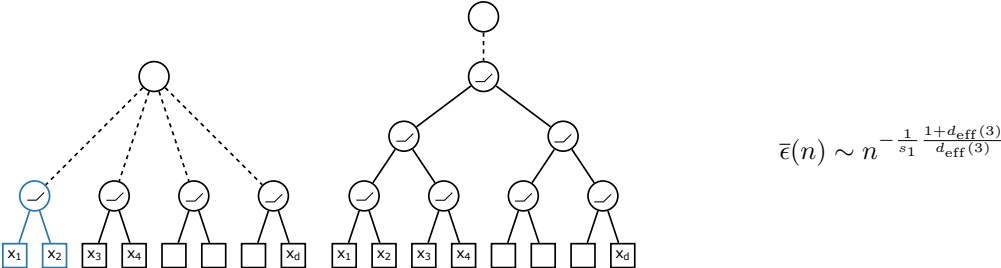

$$\bar{\epsilon}(n) \sim n^{-\frac{1}{s_1} \frac{1 + d_{\text{eff}}(3)}{d_{\text{eff}}(3)}}$$

Remarkably, this student is able to learn the teacher without being cursed by input dimensionality. Indeed, as the number of patches diverges, the error decay asymptotes to $n^{-1/s_1}$. This rate is slightly worse than the one obtained by the student matched with the teacher, which is proven to be the Bayes-optimal case, but far from being cursed. Intuitively, this fast rate is obtained because the student eigenfunctions of the first sector, i.e. constant outside a single patch, correspond to large eigenvalues and bias the learning dynamics towards $s_1$-local functions. Yet, this student is also able to represent functions which are considerably more complex.

Now consider a depth-three teacher (left) learned by a depth-four student (right).

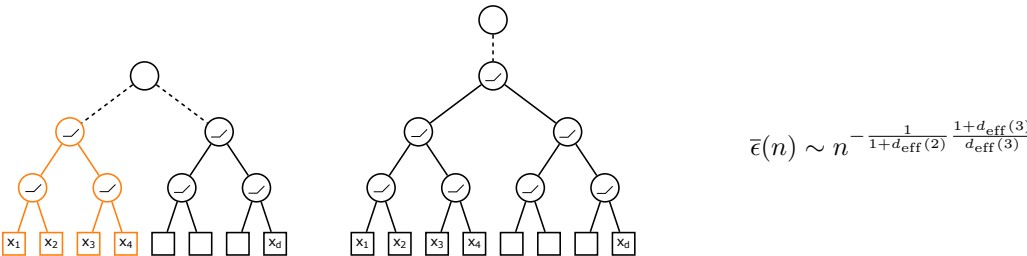

$$\bar{\epsilon}(n) \sim n^{-\frac{1}{1 + d_{\text{eff}}(2)} \frac{1 + d_{\text{eff}}(3)}{d_{\text{eff}}(3)}}$$

As highlighted in orange, the output of the teacher is a linear combination of a composition of non-linear functions acting on patches and coupling them. In this setting, the error decay is controlled by the effective dimension of the second layer. In fact, when the number of patches diverges, the error

decay asymptotes to $n^{-1/d_{\text{eff}}(2)}$. In general, this behaviour is a result of what we called 'adaptivity to the spatial structure' of the target.

Finally, consider both teacher and student with the complete hierarchy, i.e. the receptive fields of the neurons in the penultimate layers coincide with the full input.

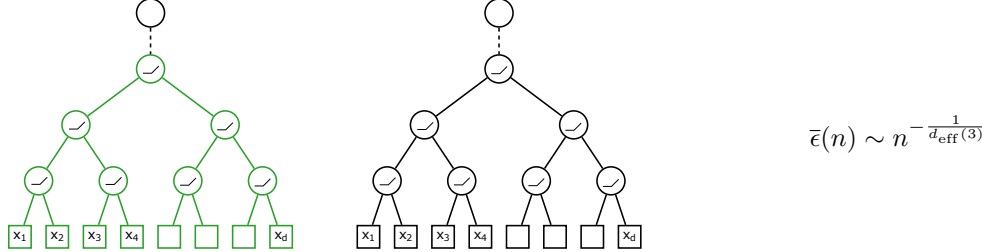

$$\bar{\epsilon}(n) \sim n^{-\frac{1}{d_{\text{eff}}(3)}}$$

In this case, we show that the error decays as $n^{-1/d_{\text{eff}}(3)}$, i.e. the rate is cursed by the input dimension. The physical meaning of this result is that the hierarchical structure we are considering is still too complex and cannot be learnt efficiently. In other words, these hierarchical convolutional networks are excellent students, since they can adapt to the spatial structure of the task, but bad teachers, since they generate global functions which are too complex to be learnt efficiently.

### G.4 EXTENSIONS TO DIFFERENT NORMALISATIONS AND OVERLAPPING PATCHES

This section investigates the robustness of our results to changes in the input distribution, i.e., for data outside the multisphere $\mathsf{M}^p\mathbb{S}^{s-1}$, and relaxes the non-overlapping patches assumption.

**Inputs in $\mathbb{R}^d$.** While our analysis requires that each patch of the input data is normalised to lie on a unit sphere, this normalisation is not the standard one used for neural networks. Therefore, in this section we investigate the robustness of our predictions to the data distribution. In particular, we consider data uniformly distributed in the unit hypercube, i.e., $\boldsymbol{x} \in [0,1]^d$, and data with standard Gaussian distribution, i.e., $\boldsymbol{x} \sim \mathcal{N}(0, I_d)$. First, we extend the definition of the RFK and NTK to inputs in $\mathbb{R}^d$.

**Definition G.1 (RFK and NTK of hierarchical CNNs for inputs in $\mathbb{R}^d$)** *Let $\boldsymbol{x}, \boldsymbol{y} \in \mathbb{R}^d$. Denote tuples of the kind $i_l i_{l+1} \ldots i_m$ with $i_{l \to m}$ for $m \geq l$. For $m < l$, $i_{l \to m}$ denotes the empty tuple. For each tuple $i_{2 \to L+1}$ and $s$ a divisor of $d$, denote with $t_{i_{2 \to L+1}}$ the angle between the $s$-dimensional patches of $\boldsymbol{x}$ and $\boldsymbol{y}$ identified by the same tuple, i.e.*

$$t_{i_{2 \to L+1}} = \frac{\boldsymbol{x}_{i_{2 \to L+1}} \cdot \boldsymbol{y}_{i_{2 \to L+1}}}{\|\boldsymbol{x}_{i_{2 \to L+1}}\| \|\boldsymbol{y}_{i_{2 \to L+1}}\|} \tag{S88}$$

*For $1 \leq l \leq L+1$, denote with $\left\{\boldsymbol{x}_{i_{2 \to L+1}}, \boldsymbol{y}_{i_{2 \to L+1}}\right\}_{i_{2 \to l}}$ the sequence of patches obtained by letting the indices of the tuple $i_{2 \to l}$ vary in their respective range. Consider a hierarchical CNN with filter sizes $(s_1, \ldots, s_L)$, $p_L \geq 1$ and all the weights $w_{h,i}^{(1)}, w_{h,h',i}^{(l)}, w_{h,i}^{(L+1)}$ initialised as Gaussian random numbers with zero mean and unit variance.*

*RFK. The corresponding RFK (or covariance kernel) can be obtained recursively as follows. With $\kappa_1(t) = \left((\pi - \arccos t)\, t + \sqrt{1 - t^2}\right)/\pi$,*

$$\mathcal{K}_{\text{RFK}}^{(1)}(\boldsymbol{x}_{i_{2 \to L+1}}, \boldsymbol{y}_{i_{2 \to L+1}}) = \|\boldsymbol{x}_{i_{2 \to L+1}}\| \|\boldsymbol{y}_{i_{2 \to L+1}}\| \, \kappa_1(t_{i_{2 \to L+1}});$$

$$\mathcal{K}_{\text{RFK}}^{(l)}\left(\left\{\boldsymbol{x}_{i_{2 \to L+1}}, \boldsymbol{y}_{i_{2 \to L+1}}\right\}_{i_{2 \to l}}\right) = \sqrt{\frac{1}{s_l} \sum_{i_l} \|\boldsymbol{x}_{i_l \to L+1}\|^2} \sqrt{\frac{1}{s_l} \sum_{i_l} \|\boldsymbol{y}_{i_l \to L+1}\|^2}$$

$$\times \kappa_1\left(\frac{\frac{1}{s_l} \sum_{i_l} \mathcal{K}_{\text{RFK}}^{(l-1)}\left(\left\{\boldsymbol{x}_{i_{2 \to L+1}}, \boldsymbol{y}_{i_{2 \to L+1}}\right\}_{i_{2 \to l-1}}\right)}{\sqrt{\frac{1}{s_l} \sum_{i_l} \|\boldsymbol{x}_{i_l \to L+1}\|^2} \sqrt{\frac{1}{s_l} \sum_{i_l} \|\boldsymbol{y}_{i_l \to L+1}\|^2}}\right);$$

$$\mathcal{K}_{\text{RFK}}^{(L+1)}\left(\left\{\boldsymbol{x}_{i_{2 \to L+1}}, \boldsymbol{y}_{i_{2 \to L+1}}\right\}_{i_{2 \to L+1}}\right) = \frac{1}{p_L} \sum_{i_{L+1}=1}^{p_L} \mathcal{K}_{\text{RFK}}^{(L)}\left(\left\{\boldsymbol{x}_{i_{2 \to L+1}}, \boldsymbol{y}_{i_{2 \to L+1}}\right\}_{i_{2 \to L}}\right). \tag{S89}$$

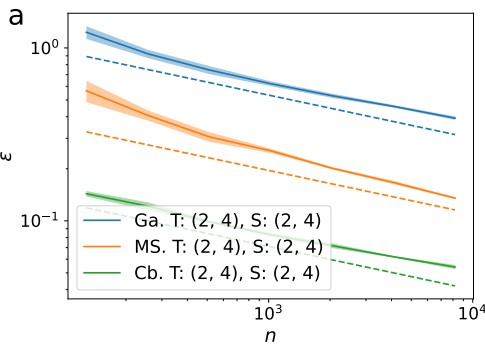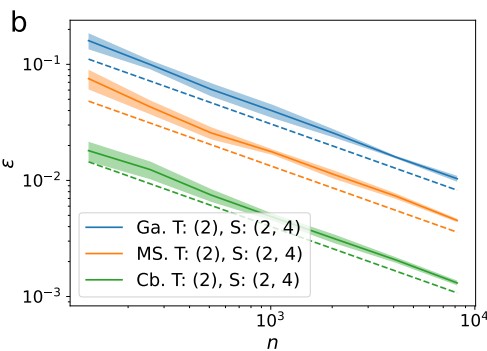

Figure S4: Learning curves for deep convolutional NTKs ($\nu = 1/2$) in a teacher-student setting with different input normalisations. In particular, we consider inputs on the multisphere $\mathsf{M}^p \mathbb{S}^{s-1}$ (MS.), uniformly-distributed in the unit $d$-hypercube $[0,1]^d$ (Cb.), and with standard Gaussian distribution $\mathcal{N}(0, I_d)$ (Ga.). The numbers inside brackets are the sequence of filter sizes of the kernels. Solid lines are the results of experiments averaged over 16 realisations with the shaded areas representing the empirical standard deviations. The asymptotic scaling $\epsilon \sim n^{-\beta}$ predicted for inputs on the multisphere are reported as dashed lines.

***NTK.*** *The NTK of the same hierarchical CNN can be obtained recursively as follows. With* $\kappa_0(t) = (\pi - \arccos t) / \pi,$

$$\mathcal{K}_{\text{NTK}}^{(1)}\left(\boldsymbol{x}_{i_{2\to L+1}}, \boldsymbol{y}_{i_{2\to L+1}}\right) = \|\boldsymbol{x}_{i_{2\to L+1}}\| \|\boldsymbol{y}_{i_{2\to L+1}}\| \kappa_1(t_{i_{2\to L+1}}) + \boldsymbol{x}_{i_{2\to L+1}} \cdot \boldsymbol{y}_{i_{2\to L+1}} \, \kappa_0(t_{i_{2\to L+1}});$$

$$\mathcal{K}_{\text{NTK}}^{(l)}\left(\left\{\boldsymbol{x}_{i_{2\to L+1}}, \boldsymbol{y}_{i_{2\to L+1}}\right\}_{i_{2\to l}}\right) = \mathcal{K}_{\text{RFK}}^{(l)}(\left\{\boldsymbol{x}_{i_{2\to L+1}}, \boldsymbol{y}_{i_{2\to L+1}}\right\}_{i_{2\to l}})$$

$$+ \left(\frac{1}{s_l}\sum_{i_l} \mathcal{K}_{\text{NTK}}^{(l-1)}\left(\left\{\boldsymbol{x}_{i_{2\to L+1}}, \boldsymbol{y}_{i_{2\to L+1}}\right\}_{i_{2\to l-1}}\right)\right)$$

$$\times \kappa_0\left(\frac{\frac{1}{s_l}\sum_{i_l} \mathcal{K}_{\text{RFK}}^{(l-1)}\left(\left\{\boldsymbol{x}_{i_{2\to L+1}}, \boldsymbol{y}_{i_{2\to L+1}}\right\}_{i_{2\to l-1}}\right)}{\sqrt{\frac{1}{s_l}\sum_{i_l}\|\boldsymbol{x}_{i_{l\to L+1}}\|^2}\sqrt{\frac{1}{s_l}\sum_{i_l}\|\boldsymbol{y}_{i_{l\to L+1}}\|^2}}\right);$$

$$\mathcal{K}_{\text{NTK}}^{(L+1)}\left(\left\{\boldsymbol{x}_{i_{2\to L+1}}, \boldsymbol{y}_{i_{2\to L+1}}\right\}_{i_{2\to L+1}}\right) = \frac{1}{p_L}\sum_{i_{L+1}=1}^{p_L} \mathcal{K}_{\text{NTK}}^{(L)}\left(\left\{\boldsymbol{x}_{i_{2\to L+1}}, \boldsymbol{y}_{i_{2\to L+1}}\right\}_{i_{2\to L}}\right). \quad \text{(S90)}$$

Fig. S4 reports the learning curve of different teacher-student scenarios with the kernels defined in Def. G.1 and inputs *i)* on the multisphere $\mathsf{M}^p \mathbb{S}^{s-1}$, *ii)* uniformly-distributed in the unit $d$-hypercube $[0,1]^d$, and *iii)* with standard Gaussian distribution $\mathcal{N}(0, I_d)$. Remarkably, our predictions are in excellent agreement with the different input normalisations.

**Overlapping patches.** Fig. S5 shows the comparison between convolutional kernels with non-overlapping patches, i.e., stride corresponding to the filter size, and overlapping patches, i.e., stride 1, for inputs uniform in the $d$-dimensional hypercube. Despite our theoretical analysis requires the patches to be non-overlapping, our predictions are still confirmed for architectures with overlapping patches.

### G.5 CIFAR-2 LEARNING CURVES

Fig. S6 shows the learning curves of the neural tangent kernels of different architectures applied to pairs of classes of the CIFAR-10 dataset. In particular, the task is built by selecting two CIFAR-10 classes, e.g. plane and car, and assigning label $+1$ to the elements belonging to one class and label $-1$ to the remaining ones. Learning is again achieved by minimising the empirical mean squared error using a 'student' kernel. We find that the kernels with the worst performance are the ones corresponding to shallow fully-connected and convolutional architectures. Instead, for all the pairs of classes considered here, deep hierarchical convolutional kernels achieve the best performance.

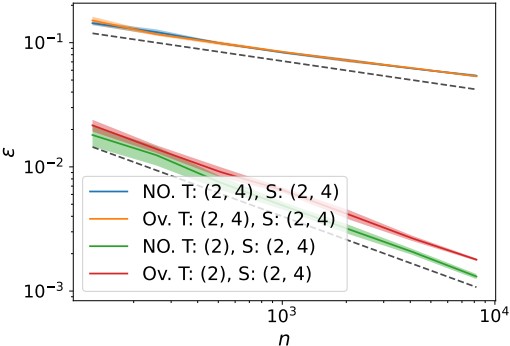

Figure S5: Learning curves for deep convolutional NTKs ($\nu = 1/2$) with non-overlapping (NO.) and overlapping (Ov.) patches in a teacher-student setting with inputs normalised in the $d$-hypercube. The numbers inside brackets are the sequence of filter sizes of the kernels. Solid lines are the results of experiments averaged over 16 realisations with the shaded areas representing the empirical standard deviations. The asymptotic scaling $\epsilon \sim n^{-\beta}$ predicted for kernels with non-overlapping patches are reported as dashed lines.

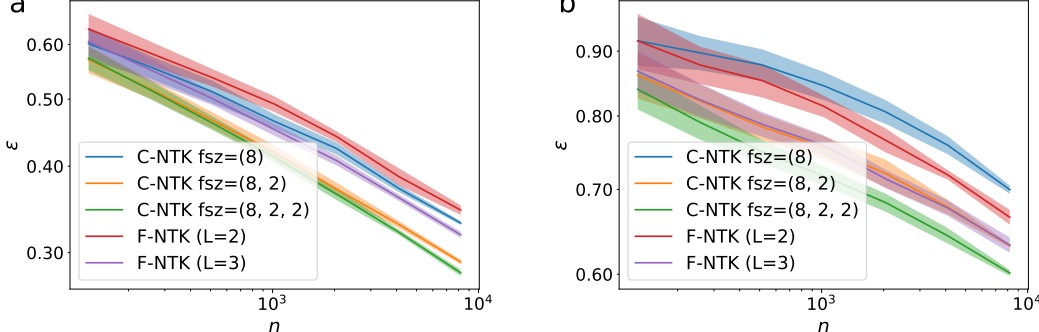

Figure S6: Learning curves of the neural tangent kernels of fully-connected (F-NTK) and convolutional (C-NTK) networks with various depths learning to classify two CIFAR-10 classes in a regression setting. Deep hierarchical convolutional kernels achieve the best performance. Shaded areas represent the empirical standard deviations obtained averaging over different training sets. **a.** Plane vs car. **b.** Cat vs bird.

