# OpenReview forum: "What can be learnt with wide convolutional neural networks?"
_ICLR.cc/2023/Conference — Submitted to ICLR 2023_

### Official Review · Reviewer_gyaw · 2022-10-21

**Confidence:** 2
**Correctness:** 3
**Technical Novelty And Significance:** 3
**Empirical Novelty And Significance:** 2
**Recommendation:** 6

**Clarity, Quality, Novelty And Reproducibility:**

I cannot fully cover but the theoretical derivation looks correct.

The quality of the paper is high. Significant theoretical results are provided. Small but sufficient experiments are also provided.

I'm not working on this area and I cannot judge the novelty.

The reproducibility is OK. Although the code is not provided, fine details of experiments (learning rate, etc) are shown in Appendix.

**Strength And Weaknesses:**

Strengths
1. The theoretical results are interesting. They clarify why CNNs are efficient in terms of learning theory.
1. The experiments provide additional evidence that the decay rate is correct.

Weaknesses
1. The assumption of Eq 1 looks unrealistic. It poses that the input space should be the product of spheres, such as a torus. Images, a favorite data of CNNs, are not in that kind of space --- they are usually represented in a cube [0, 1]^{height x width x channel}. When an input is in a spherical space, it should be steerable and the same input should be obtained by steering 360 degrees. This property doesn't hold in the raw image space. I wonder how this assumption is critical in your analysis. Can we obtain the same result without the assumption? Also, does the assumption hold in the experiments?


**Summary Of The Paper:**

This paper studies how the generalization error decreases in multilayer CNNs. Using NTK, the authors theoretically show that the decay rate of the error is determined by the effective dimension of the data instead of the actual dimension. Here the effective dimension relates to how each hidden unit _sparsely_ depends on the output of the previous layer. The experiments also support the derived decay rate.

**Summary Of The Review:**

The paper brings a neat analysis of CNN in terms of generalization error. My concern is that the assumption used in the analysis do not seem to match real problems.

---

> ### Author Response · Authors · 2022-11-10
> **Answer to Reviewer gyaw**
>
> Let us thank the reviewer for recognising the strengths of our work. Let us also mention that we have uploaded a revised version of the manuscript together with a zip with the code used for the experiments (a normal link will be included in the final version if the paper is accepted).
>
> Our choice of input space is required in order to use tools from harmonic analysis when characterising the kernel spectrum. The reviewer considers this choice a limitation. We argue that it is not a significant limitation for the following reasons.
>
> - The normalisation of the inputs is known in the field as "unit vector normalisation" or "spatial sign preprocessing" and it is often used in practice in the kernel literature (see, e.g., `https://scikit-learn.org/stable/modules/preprocessing.html#normalization`). In our work, we apply this normalisation on all the patches rather than on the whole image. All our experiments use this normalisation, including those on CIFAR.
>
> - In addition, our results are robust to changes in normalisation. We have performed additional experiments with inputs taken from the hypercube suggested by the reviewer, and also with points taken from a Gaussian distribution in $\mathbb{R}^d$. These experiments show that our predictions on the error decay hold irrespectively of these choices. We now highlight this point in the revised version of the manuscript and included the aforementioned experiments in Appendix G.
>
> Finally, we respectfully disagree with the reviewer when they say that "this property (steerability) doesn't hold in the raw image space". While it is true that, once an image is represented as a vector of arbitrary dimension $d$, a finite rotation might completely destroy the image content, any 360-degree rotation coincides *by definition* with the identity, thus it cannot have any impact on the image.
>
> We hope to have addressed the remark and thank the reviewer again for the feedback.

---

### Official Review · Reviewer_UzrZ · 2022-10-24

**Confidence:** 2
**Correctness:** 4
**Technical Novelty And Significance:** 3
**Empirical Novelty And Significance:** 3
**Recommendation:** 6

**Clarity, Quality, Novelty And Reproducibility:**

This paper is well written. However, the organization may need to be improved since I find it quite hard to follow.

**Strength And Weaknesses:**

Strength:
1. The idea is novel.
2. This paper is comprehensive, with the solid support of proof and experiment.

Weakness:

1. This paper is hard to follow since the claims are scattered in the paper, which is hard to find.

2. This paper only cares about the NN kernel solution. So it doesn't have any optimization result or discussion of the width.

3. This paper requires patch non-overlapping. It is unclear whether this paper can be extended to a more general setting.



**Summary Of The Paper:**

This paper studies convolutional neural networks in the kernel regime. In particular, they show that the corresponding kernel's spectrum inherits the network's hierarchical structure.  Therefore, deep CNNs can beat the curse of dimensionality if the target depends only on local groups of variables.

**Summary Of The Review:**

Overall, the results proved in this paper may be significant for people who study NTK. This paper feels complete, but I still have concerns about clarity.

In the abstract, the author gives the following claims. "the rate of decay of the error is controlled by the effective dimensionality of these subsets," "if the teacher function depends on the full set of input variables, then the error rate is inversely proportional to the input dimension," "despite their hierarchical structure, the functions generated by deep CNNs are too rich to be efficiently learnable in high dimension."  Where can I find the support theorems of these claims? I tried my best to understand them, but I could only find some relevant explanations rather than solid theorems.

---

> ### Author Response · Authors · 2022-11-10
> **Answer to Reviewer UzrZ**
>
> Let us thank the reviewer for recognising the comprehensiveness of our work and for providing us with very well-structured comments. We will reply to the reviewer's remarks point by point.
>
>
> **1.** "hard to follow and claims are scattered"
>
> The reviewer seems to refer specifically to some claims given in the abstract. All these claims are explained in Section 5 of the original manuscript (first paragraph for the error rate controlled by the effective dimensionality, second paragraph for the error rate of global functions depending on the inverse input dimension, and last paragraph for the functions generated by deep CNNS not being learnable in high dimension). We agree with the reviewer that these results should be highlighted and organised better and we have changed the manuscript accordingly.
>
> The revised manuscript contains a modified version of Corollary 4.1. Now, we clarify why the error decay rate is controlled by the effective dimensionality of the target, and thus the full dimensionality if the target depends on all the input variables.
>
> The point about functions generated by deep CNNs not being learnable requires two additional steps in addition to Corollary 4.1:
>
> - a Gaussian random function with covariance kernel given by the random feature kernel is equivalent to an infinitely-wide network with Gaussian initialisation of the weights;
>
> - the optimal method (in the sense of Bayes) to learn such function is a kernel method with a kernel which coincides with the function covariance. To clarify this point, we have turned the last paragraph of Section 5 into a lemma where the claim is laid out in detail.
>
> We have also introduced minor modifications to the section "Our contributions" to point out these additions.
>
>
> **2.** "no optimization results or discussion of the width"
>
> The solution studied in our work is provably reached by gradient-descent optimisation in the infinite-width limit. The reviewer is correct that our results do not apply to the finite-width case where features are learnt. Treating the feature learning case in general for deep architectures is a major challenge in the field. Arguably, understanding generalisation in the kernel regime is a necessary first step to solving this challenge. Indeed our work already unveils some critical advantages of deep convolutional architectures over shallow and/or fully-connected ones, such as adaptivity to the spatial structure.
>
>
> **3.** "requirement of nonoverlapping patches"
>
> This assumption is indeed required for us to be able to calculate the kernel spectrum. However, it is technical in nature. To support this claim we have performed additional experiments with different input spaces (see also reply to reviewer gyaw) and with kernels having overlapping patches. The results are presented in a new subsection of Appendix G. They show that the general predictions of our theory also apply to the overlapping-patches case.

---

### Official Review · Reviewer_xmg3 · 2022-10-26

**Confidence:** 4
**Correctness:** 4
**Technical Novelty And Significance:** 2
**Empirical Novelty And Significance:** 1
**Recommendation:** 3

**Clarity, Quality, Novelty And Reproducibility:**

The paper is well written in general. But the network structure based on a simply average is too special and prevents applications of the derived analysis to network designs.

**Strength And Weaknesses:**

Deep CNNs have been studied theoretically recently and generalization error analysis has been carried out for CNNs of various types. The spectral and convergence analysis provided in the paper is interesting. But the network structure in Equation (1) uses a special linear transformation of a simply average over h', instead of a general linear combination, which is very special. This restriction makes the mathematical analysis easier. More discussion on networks with a general linear transformation would increase the scientific value of the paper.

**Summary Of The Paper:**

The authors provide some generalization analysis for deep CNNs. Based on some spectral analysis, they prove that deep CNNs adapt to the spatial scale of the target function. They also show how the error rate depends on the input dimension.

**Summary Of The Review:**

The spectral and convergence analysis provided in the paper for deep CNNs is interesting. But the network structure in Equation (1) uses a special linear transformation. This restriction makes the mathematical analysis easier. More discussion on general networks would enhance the paper.

---

> ### Author Response · Authors · 2022-11-10
> **Answer to Reviewer xmg3**
>
> We thank the reviewer for appreciating our spectral analysis. The main criticism of the reviewer regards the network structure, but we believe it is due to a misunderstanding. The layers of our hierarchical networks (Eq. (1)) perform a generic linear combination of their input and not an average (note that the weights of the $l$-th hidden layer have two indices, $h$ and $h'$). This operation is equivalent to that performed by the standard generic convolutional layers of PyTorch, `conv1d` and `conv2d`.
>
> The only simplification with respect to the generic case, as pointed out by reviewer UzrZ, is the nonoverlapping patches assumption, which requires the stride of each layer to be equal to the filter size. As discussed below (see reply to reviewer UzrZ), this assumption is only technical: in the revised manuscript, we verify empirically that our predictions still hold in the general case. We hope to have addressed the reviewer's criticism and we will be happy to provide further clarifications if required.

---

> ### Author Response · Authors · 2022-12-06
> **Message to Reviewer xmg3**
>
> Dear Reviewer xmg3,
>
> As the review period is approaching its end, we hope you have had a chance to read our previous rebuttal and consider our concerns. As we already commented, we seriously believe that your recommendation is due to a misunderstanding stemming from an erroneous interpretation of the structure of our architecture. Therefore, we would like to request that you reconsider your score.
>
> Thank you for reviewing our work.
>
> Sincerely,
>
> The Authors

---

### Author Response · Authors · 2022-11-10
**Message to All Reviewers**

We thank all reviewers for their feedback on our manuscript. We point out that we updated the paper taking into account the comments. All modifications are highlighted in blue. We hope to have addressed all your questions and we remain available to provide further clarifications.

---

### Decision · Program_Chairs · 2023-01-20

**Decision:**

Reject

**Justification For Why Not Higher Score:**

Fairly consistent reviewer concerns

**Justification For Why Not Lower Score:**

N/A

**Metareview: Summary, Strengths And Weaknesses:**

This paper studies the kernel limit of certain families of CNNs, showing a variety of interesting results.  Unfortunately, as detailed in the individual reviews, there were a variety of simplifying assumptions and presentations of the work which detracted from this message.  As such, I recommend the authors continue their exciting work, address the reviewer concerns, and submit to a future venue.